# FlowAD: Ego-Scene Interactive Modeling for Autonomous Driving

**Mingzhe Guo**[1]     **Yixiang Yang**[2]     **Chuanrong Han**[2]     **Rufeng Zhang**[1]

**Shirui Li**[1]     **Ji Wan**[1]     **Zhipeng Zhang**[2*]

[1]Baidu Inc.     [2]AutoLab, School of Artificial Intelligence, Shanghai Jiao Tong University

## Abstract

Effective environment modeling is the foundation for autonomous driving, underpinning tasks from perception to planning. However, current paradigms often inadequately consider the feedback of ego motion to the observation, which leads to an incomplete understanding of the driving process and consequently limits the planning capability. To address this issue, we introduce a novel ego-scene interactive modeling paradigm. Inspired by human recognition, the paradigm represents ego-scene interaction as the scene flow relative to the ego-vehicle. This conceptualization allows for modeling ego-motion feedback within a feature learning pattern, advantageously utilizing existing log-replay datasets rather than relying on scenario simulations. We specifically propose FlowAD, a general flow-based framework for autonomous driving. Within it, an ego-guided scene partition first constructs basic flow units to quantify scene flow. The ego-vehicle's forward direction and steering velocity directly shape the partition, which reflects ego motion. Then, based on flow units, spatial and temporal flow predictions are performed to model dynamics of scene flow, encompassing both spatial displacement and temporal variation. The final task-aware enhancement exploits learned spatio-temporal flow dynamics to benefit diverse tasks through object and region-level strategies. We also propose a novel Frames before Correct Planning (FCP) metric to assess the scene understanding capability. Experiments in both open and closed-loop evaluations demonstrate FlowAD's generality and effectiveness across perception, end-to-end planning, and VLM analysis. Notably, FlowAD reduces 19% collision rate over SparseDrive with FCP improvements of 1.39 frames (60%) on nuScenes, and achieves an impressive driving score of 51.77 on Bench2Drive, proving the superiority. Code, model, and configurations will be released here .

## 1 Introduction

Autonomous driving has achieved remarkable progress in recent decades Ly & Akhloufi (2020); Kiran et al. (2021). A pivotal development is the shift from modular designs, with discrete perception, prediction and planning stages Zhu & Zhao (2021); Hussein et al. (2017), to end-to-end (E2E) architectures Sadat et al. (2020); Hu et al. (2023). This transition optimizes the flow of planning-centric information, minimizing intermediate losses and thereby significantly enhancing the performance. More recently, the escalating need for generalized scene understanding and reasoning has driven the adoption of Large Vision-Language Models (LVLMs) within these systems Tian et al. (2024); Jiang et al. (2024). The integration unlocks further potential in data-driven learning, endowing it with improved capabilities for holistic scene comprehension and more human-like ego-vehicle planning.

Upon reviewing existing architectures, it is evident that the planning module consistently operates as the final computational step, contingent upon environmental messages from preceding modules. Each inference cycle culminates in an ego-plan, after which the pipeline resets for the next timestamp (Fig. 1a). Critically, however, such architecture largely neglects the profound impact of the ego-vehicle's own executed motion on its subsequent perception and decision-making. A complete

---

*Correspondence Author

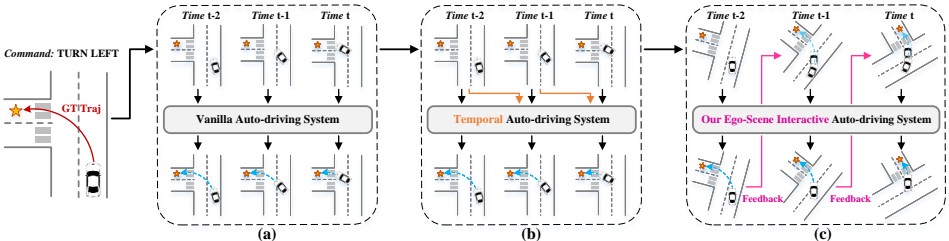

Figure 1: **(a)** Vanilla auto-driving system that performs isolated inference for each timestamp. **(b)** Temporal auto-driving system that integrates historical observations, yet incompletely captures the feedback of previous ego planning. **(c)** Our ego-scene interactive system that leverages previous planning to inform future observations, which benefits comprehension of dynamic driving process.

Table 1: Influence of the temporal fusion in UniAD Hu et al. (2023) on the nuScenes validation set.

| Method | Backbone | Detection | | Tracking | | Online Map | | Motion Prediction | | Planning | |
|---|---|---|---|---|---|---|---|---|---|---|---|
| | | mAP↑ | NDS↑ | AMOTA↑ | AMOTP↓ | IoU-Lane↑ | IoU-Road↑ | minADE↓ | minFDE↓ | Avg.L2↓ | Avg.Col↓ |
| UniAD Hu et al. (2023) | ResNet101 | 0.380 | 0.498 | 0.359 | 1.320 | 0.302 | 0.672 | 0.71 | 1.02 | 1.03 | 0.31 |
| UniAD w/o.Tem Hu et al. (2023) | ResNet101 | 0.356 *-6%* | 0.473 *-5%* | 0.303 *-16%* | 1.549 *-17%* | 0.288 *-5%* | 0.648 *-4%* | 0.86 *-21%* | 1.22 *-20%* | 1.08 *-5%* | 0.33 *-6%* |

driving process should involve two parts: planning with current observation, and more importantly, performing the control outputs that shape future sensory inputs. The absence of the second part, *i.e.,* the feedback of ego motion, leads to different open- and closed-loop environments Caesar et al. (2020); Dosovitskiy et al. (2017). The closed-loop environment Jia et al. (2024) is mainly used for evaluation with real-time interactions between the ego vehicle and driving scene, hence allowing for more realistic testing of driving policies. In contrast, the open-loop one Contributors (2024) without ego-motion feedback is applied for large-scale training and testing of most current auto-driving systems due to its simplicity. With the fixed, pre-recorded datasets, planned trajectories are not physically realized, thereby severing the link between action and subsequent observation. This decoupling significantly impedes the model's ability to internalize the complex, dynamic interplay inherent in ego-scene interactions, ultimately constraining its planning acumen. Even temporal architecture (Fig. 1b) integrating historical states to model environmental changes, it often fails to fully capture the nuanced feedback from ego-actions to future states. The findings in Tab. 1 underscore this point, that removing temporal fusion has a negligible effect on planning, yet severely hampers tasks like tracking that depend on temporal continuity, suggesting that current temporal modeling does not adequately address this ego-centric feedback loop for planning.

To address the aforementioned limitation, we introduce a novel ego-scene interactive modeling architecture. This architecture is designed to explicitly incorporate the feedback of the ego-vehicle's motion by learning its influence within the latent feature space. As illustrated in Fig. 1c, our approach leverages the planned ego trajectory from the preceding timestep to inform the reconstruction or prediction of subsequent environmental observations, thereby explicitly modeling the dynamic interplay between the ego-vehicle and its surroundings. The core intuition for this methodology is drawn from human perceptual-motor processes, particularly the concept of relative motion Davis & Bobick (1997); Bobick & Davis (2001). Humans inherently understand that as they move, the environment appears to flow in the opposite direction. This perceived optic flow is crucial for anticipatory planning and navigation. We posit that this fundamental aspect of ego-scene interaction, manifested as relative motion, can be effectively captured and represented as a learnable "scene flow" within the latent space of the model. Crucially, this formulation enables the modeling of ego-motion feedback using readily available, pre-recorded datasets, obviating the need for complex simulations to generate varied observational outcomes.

More specifically, we introduce FlowAD, a general flow-based framework for auto-driving, structured around three core components: **1) Ego-guided scene partition** addresses the inherent difficulty in quantifying holistic scene flow by decomposing the visual input into "flow units". This partitioning is strategically performed along the width of multi-view images, aligning with the predominant horizontal ego-scene relative motion, and its starting point and unit granularity dynamically adapt to the ego-vehicle's forward direction and steering velocity. To mitigate potential object fragmentation and capture scene flows across varied receptive fields, both local aggregation and

multi-level partition strategies are integrated. **2) Spatial and temporal flow prediction** leverages these flow units to model the scene flow dynamics of spatial displacement and temporal variation. A spatial module infers the states of subsequent units based on information from antecedent ones, while a temporal module forecasts future unit states by exploiting historical ones. Mechanisms from world models Hafner et al. (2020; 2021) are incorporated for refined modeling of flow dynamics. **3) Task-aware enhancement** exploits the learned spatio-temporal flow dynamics to improve diverse downstream tasks. Specifically, object queries for object-level perception are augmented using information from corresponding flow units, and region features for region-level analysis are boosted. This enhanced understanding of scene dynamics facilitates more agile system responses.

Experiments in perception, open-/closed-loop E2E planning, and VLM analysis validate FlowAD's effectiveness and generality. Applied to baselines SparseBEV Liu et al. (2023), SparseDrive Sun et al. (2024), and Senna Jiang et al. (2024), our framework yields consistent performance gains. To assess the scene understanding capability, we also introduce a novel Frames before Correct Planning (FCP) metric, which quantifies the frames elapsed until the planner initiates a rational action in response to a given command.

In summary, our contributions are three-fold: **1)** We observe the limited planning capability caused by neglecting the feedback of ego motion, and propose an ego-scene interactive modeling paradigm to build the influence by learning the scene flow in latent space; **2)** We design a general flow-based framework for autonomous driving, which builds and exploits the flow dynamics to benefit diverse downstream tasks; **3)** We prove the effectiveness and generality of our method with SOTA performance in both open- and closed-loop evaluations, as well as the proposed FCP metric.

## 2 RELATED WORK

### 2.1 END-TO-END AUTONOMOUS DRIVING

In contrast to the traditional auto-driving systems that perform staged and rule-based planning Zhu & Zhao (2021), the end-to-end architecture Hu et al. (2023) directly outputs the planning results with the input environmental observation. From the seminal work ALVINN Pomerleau (1988) to modern approaches such as ST-P3 Hu et al. (2022b), dedicated modular designs are introduced to integrate auxiliary information. UniAD Hu et al. (2023) and VAD Jiang et al. (2023) demonstrate impressive performance with Transformer pipelines. After that, more progress has been achieved, *e.g.,* sparse modeling in SparseDrive Sun et al. (2024), probabilistic planning in VADv2 Chen et al. (2024). Most recently, the capabilities of commonsense reasoning and interoperability in LVLMs are employed. The pioneering DriveGPT4 Xu et al. (2024) converts sensor inputs into control commands via VLMs. Building on hybrid designs, DriveVLM Tian et al. (2024) and Senna Jiang et al. (2024) integrate LVLMs with planners to achieve unified high-level reasoning and low-level control. In this work, we further consider the feedback of ego motion and model ego-scene interactive dynamics to benefit the driving comprehension.

### 2.2 WORLD MODEL

With the formulated representation of the world, the world models Ha & Schmidhuber (2018b;a) aim to learn the transition dynamics and predict the future states. The studies in either gaming or robotics are widely explored Hafner et al. (2020; 2021; 2023). Recently, learning world models have been introduced into autonomous driving Hu et al. (2022a); Wang et al. (2024b;a). MILE Hu et al. (2022a) incorporates world modeling within the BEV segmentation through imitation learning. The successive DriveDreamer Wang et al. (2024a), ADriver-I Jia et al. (2023) and Drive-WM Wang et al. (2024b) explore the training of driving world models with diffusion designs. In this work, we employ the mechanism of the world model to learn spatio-temporal dynamics of the scene in the latent space, which helps to understand the ego-scene interaction and improve the planning capability.

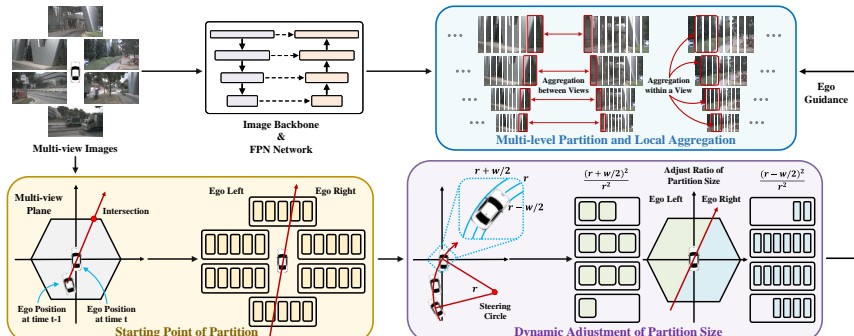

Figure 2: Illustration of Ego-guided Scene Partition. The feedback of ego motion is reflected in Starting Point of Partition (forward direction) and Dynamic Adjustment of Partition Size (velocity). Then Multi-level Partition and Local Aggregation divides basic flow units and fuses local messages.

## 3 METHOD

### 3.1 EGO-GUIDED SCENE PARTITION

The ego-guided scene partition aims to construct the basic units to model the ego-scene interactive dynamics based on the ego motion and input multi-view images, as in Fig. 2. The partition is performed on the multi-view image features $\mathbf{F}_{img} \in \mathbb{R}^{N \times H \times W \times C}$ built with the backbone network.

**Starting Point of Partition.** The ego-vehicle's forward direction determines the start from which the driving scenario flows through relative motion. Thus, we first consider the starting point of partition to introduce the guidance of ego motion. As shown in Fig. 2, we assume the ego-vehicle at time $t$ is located at the origin of the coordinate system, with six camera planes arranged at the edges of the perception range. The ego-vehicle's forward direction is formed as a vector (red arrow), constructed with ego positions at time $t-1/t$. Then the intersection (red point) between the forward vector and multi-view planes serves as the start of partition, which divides the ego-left/right scenes.

**Dynamic Adjustment of Partition Size.** During steering maneuvers, the flow speeds of the ego-vehicle's left/right scenes vary due to different lateral movement velocities. In such cases, partitioning the bilateral scenes with the same size $P$ does not conform to the kinematic characteristics of ego-scene interaction. We then design a strategy to dynamically adjust the partition size as illustrated in Fig. 2. Assuming the steering trajectory of the ego-vehicle is part of a circle Wang et al. (2005); Park et al. (2015), the ego positions $\{(x_{t-2}, y_{t-2}), (x_{t-1}, y_{t-1}), (x_t, y_t)\}$ are adopted to determine the center $(x_c, y_c)$ and radius $r$ (please refer to appendix for details). With the ego-vehicle width $w_{ego}$, the steering radiuses of the ego-left/right motion are achieved. As shown in the turning-right case of Fig. 2, the values are $r + \frac{w_{ego}}{2} / r - \frac{w_{ego}}{2}$, respectively. We posit that the original partition size $P$ corresponds to the steering radius $r$, then the size for the ego-left is $P_{left} = P \times \frac{(r+w_{ego}/2)^2}{r^2}$, and $P_{right} = P \times \frac{(r-w_{ego}/2)^2}{r^2}$ for the ego-right. The varied partition sizes endow more fine-grained ego-scene interactive modeling, which helps to understand the feedback of ego motion to the environmental observation.

**Multi-level Partition and Local Aggregation.** As the relative motion of the ego-vehicle to the scene is mainly reflected in the horizontal dimension, we perform partition along the width of multi-view images. As shown in Fig. 2, each image feature $\mathbf{F}^i_{img} \in \mathbb{R}^{H \times W \times C} (1 \leq i \leq N)$ is divided into multiple items, *i.e.*, flow units $\mathbf{F}_{unit} \in \mathbb{R}^{K \times H \times P \times C}$, with the predefined partition size $P$ ($W = K \times P$). The multi-level features $\{\mathbf{F}^l_{img}|1 \leq l \leq L\}$ from the backbone network are exploited to explore the flow dynamics of different receptive fields with varied partition sizes $\{P^l|1 \leq l \leq L\}$. Notably, the objects may be unexpectedly split and induce fragmentary information. We thus design local aggregation to handle this issue. The messages of adjacent flow units are fused, which helps to increase the receptive field as well as improve the correlation between multi-view images. Specifically, each flow unit $\mathbf{f}^k_{unit} \in \mathbb{R}^{H \times P \times C} (1 \leq k \leq K)$ of $\mathbf{F}_{unit}$ is concatenated with adjacent

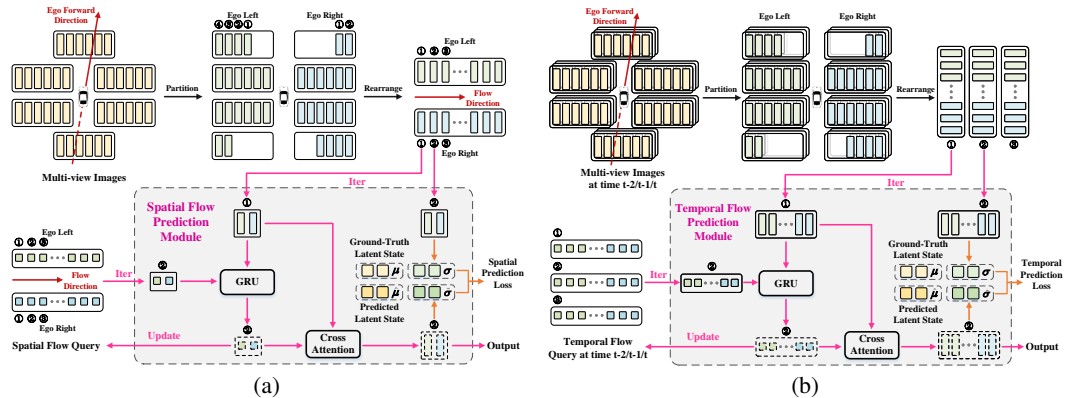

Figure 3: **(a)** The pipeline of Spatial Flow Prediction. **(b)** The pipeline of Temporal Flow Prediction.

two flow units (Fig. 2), which forms the local flow feature $\mathbf{f}_{unit}^{k-1:k+1} \in \mathbb{R}^{H \times 3P \times C}$. Then, a self-attention Vaswani et al. (2017) is performed on the $3P$ dimension to fuse local messages, followed by linear layers to reduce the dimension and output $\tilde{\mathbf{f}}_{unit}^k \in \mathbb{R}^{H \times P \times C}$. The process is formed as,

$$\tilde{\mathbf{f}}_{unit}^k = \text{MLP}(\text{SelfAttention}(\mathbf{f}_{unit}^{k-1:k+1})). \tag{1}$$

The flow units $\tilde{\mathbf{F}}_{unit} \in \mathbb{R}^{N \times K \times H \times P \times C}$ after local aggregation are input to the successive spatial and temporal flow prediction modules to build the ego-scene interactive dynamics.

### 3.2 SPATIAL AND TEMPORAL FLOW PREDICTION

With the flow units $\tilde{\mathbf{F}}_{unit}$ after ego-guided scene partition, it's capable of quantifying the scene flow of the ego-scene relative motion. The scene flow includes: 1) **spatial displacement:** the scene moves from one flow unit to another; 2) **temporal variation:** the scene of the same flow unit changes over time. Therefore, we propose spatial and temporal flow prediction modules to capture the ego-scene interactive dynamics in latent space.

**Spatial Flow Prediction Module.** We propose the spatial flow prediction module to learn the spatial dynamics of the ego-surround driving scenario. As shown in Fig. 3a, the module captures the dynamics from ahead flow units and forecasts the rear ones, which endows the capability of spatial flow prediction by supervision with GT states. Specifically, trainable spatial flow queries $\mathbf{Q}_{spat} \in \mathbb{R}^{N \times K \times C}$, which represent transition dynamics of flow units, are initialized. As the scene flows on both sides of the ego-vehicle, the flow units and queries are arranged into two parts from the partition start, *i.e.,* $\tilde{\mathbf{F}}_{unit} \in \mathbb{R}^{2 \times \frac{NK}{2} \times H \times P \times C}$ and $\mathbf{Q}_{spat} \in \mathbb{R}^{2 \times \frac{NK}{2} \times C}$. For each $\mathbf{q}_{spat}^j \in \mathbb{R}^{2 \times C}$ in $\mathbf{Q}_{spat}$, the ahead flow unit $\tilde{\mathbf{f}}_{unit}^{j-1} \in \mathbb{R}^{2 \times H \times P \times C}$ in $\tilde{\mathbf{F}}_{unit}$ is exploited to auto-regressively update the cached motional messages through a gated recurrent unit (GRU) Chung et al. (2014),

$$\hat{\mathbf{q}}_{spat}^j = \text{GRU}(\mathbf{q}_{spat}^j, \tilde{\mathbf{f}}_{unit}^{j-1}). \tag{2}$$

Notably, the ahead flow unit $\tilde{\mathbf{f}}_{unit}^0$ of the first spatial flow query $\mathbf{q}_{spat}^1$ does not exist. Thus, we take the first flow unit of the last frame $\tilde{\mathbf{f}}_{unit}^{1,t-1}$ as the substitute. As the spatial flow prediction aims to capture the scene flow dynamics, it is a prerequisite to infer the states of subsequent flow units. The output $\hat{\mathbf{q}}_{spat}^j$ is exploited to predict the rear flow unit $\hat{\mathbf{f}}_{unit}^j$ based on $\tilde{\mathbf{f}}_{unit}^{j-1}$ with a cross-attention,

$$\hat{\mathbf{f}}_{unit}^j = \text{CrossAttention}(q = \tilde{\mathbf{f}}_{unit}^{j-1}, kv = \hat{\mathbf{q}}_{spat}^j). \tag{3}$$

After iterations of all flow units, the predicted units from Eq. 3 are arranged to the spatial flow feature $\hat{\mathbf{F}}_{spat} \in \mathbb{R}^{NK \times H \times P \times C}$, which represents spatial dynamics of the scene flow.

Following the loss design in world models Hafner et al. (2020; 2021), we respectively map each predicted/GT flow units $\hat{\mathbf{f}}_{unit}^j / \tilde{\mathbf{f}}_{unit}^j$ to latent states of $\{\hat{\mu}_{spat}^j, \hat{\sigma}_{spat}^j | \mu_{spat}^j, \sigma_{spat}^j \in \mathbb{R}^{2 \times H \times P \times C'}\}$ with MLP layers, and then minimize their KL divergence. The predicted state from $\hat{\mathbf{f}}_{unit}^j$ is regarded

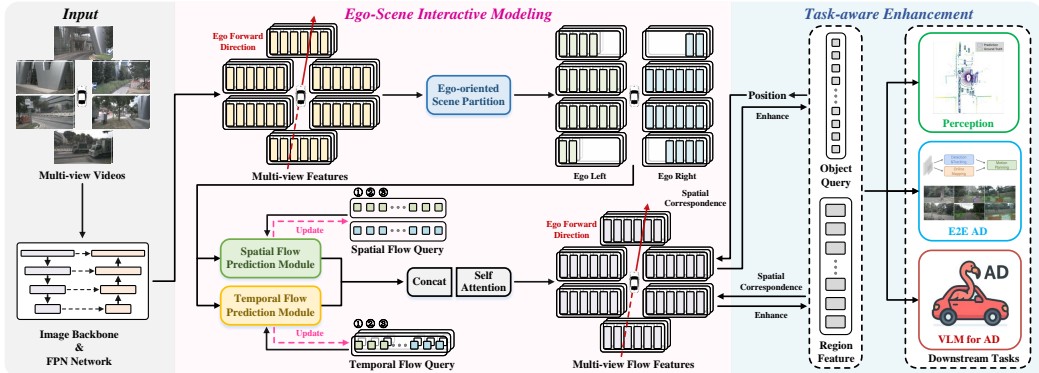

Figure 4: The architecture of our FlowAD. For the input stage, the image features of multi-view videos are extracted with the backbone network. Then, the ego-scene interactive modeling introduces the feedback of the ego-motion and builds the spatio-temporal scene flow feature. Finally, the flow feature serves for the downstream tasks with the task-aware enhancement.

as a forecast of the spatial dynamics. And the GT state from $\tilde{\mathbf{f}}_{unit}^j$ represents the distribution from the real observation. The KL divergence measures their gap. We expect to enhance the cognition ability of spatial flow by optimizing the spatial prediction loss $\mathcal{L}_{spat}$,

$$\mathcal{L}_{spat} = \mathrm{KL}(\{\hat{\mu}_{spat}^j, \hat{\sigma}_{spat}^j\}||\{\mu_{spat}^j, \sigma_{spat}^j\}). \tag{4}$$

**Temporal Flow Prediction Module.** The temporal variation of the scene flow is modeled by the proposed temporal flow prediction module. The pipeline is illustrated in Fig. 3b. Different from the spatial flow prediction within each flow unit of a single frame, the temporal flow is forecasted through a sequence of multi-view images $\{\mathbf{F}_{img}^t \in \mathbb{R}^{N \times H \times W \times C} | 1 \le t \le T\}$ with trainable temporal flow queries $\{\mathbf{Q}_{tem}^t \in \mathbb{R}^{N \times K \times C} | 1 \le t \le T\}$. For each iteration, the flow units at time $t-1$, *i.e.,* $\tilde{\mathbf{F}}_{unit}^{t-1}$, are exploited to provide temporal prior for updating the flow query $\mathbf{Q}_{tem}^t$ with a GRU,

$$\hat{\mathbf{Q}}_{tem}^t = \mathrm{GRU}(\mathbf{Q}_{tem}^t, \tilde{\mathbf{F}}_{unit}^{t-1}). \tag{5}$$

Similarly, the updated temporal flow query $\hat{\mathbf{Q}}_{tem}^t$ is exploited to predict the flow units of the next frame $\hat{\mathbf{F}}_{unit}^t$ based on $\tilde{\mathbf{F}}_{unit}^{t-1}$ with a cross-attention module,

$$\hat{\mathbf{F}}_{unit}^t = \mathrm{CrossAttention}(q = \tilde{\mathbf{F}}_{unit}^{t-1}, kv = \hat{\mathbf{Q}}_{tem}^t). \tag{6}$$

The output temporal flow feature of the last iteration, *i.e.,* $\hat{\mathbf{F}}_{unit}^T$, is employed for the downstream tasks, which provides temporal dynamics of the ego-scene interaction. The temporal prediction loss $\mathcal{L}_{tem}$ is computed with the predicted state $\{\hat{\mu}_{tem}^t, \hat{\sigma}_{tem}^t \in \mathbb{R}^{NK \times H \times P \times C'}\}$ from $\hat{\mathbf{F}}_{unit}^t$ and GT state $\{\mu_{tem}^t, \sigma_{tem}^t \in \mathbb{R}^{NK \times H \times P \times C'}\}$ from $\tilde{\mathbf{F}}_{unit}^t$ mapped with MLP layers,

$$\mathcal{L}_{tem} = \mathrm{KL}(\{\hat{\mu}_{tem}^t, \hat{\sigma}_{tem}^t\}||\{\mu_{tem}^t, \sigma_{tem}^t\}). \tag{7}$$

**Flow Feature Fusion.** After the spatial and temporal flow predictions, the output $\hat{\mathbf{F}}_{spat}$ and $\hat{\mathbf{F}}_{tem}^T$ are concatenated by flow units and fused with a self-attention, which forms $\hat{\mathbf{F}}_{fuse} \in \mathbb{R}^{NK \times H \times P \times C}$.

### 3.3 GENERAL FLOW-BASED FRAMEWORK

With the ego-guided scene partition and the spatial/temporal flow predictions, the spatio-temporal dynamics of the ego-scene interaction are represented as $\hat{\mathbf{F}}_{fuse}$. This enables the auto-driving system to understand the feedback of ego motion to the surrounding scenario, which benefits downstream tasks. We then propose a general flow-based framework for perception, end-to-end planning and VLM analysis, as shown in Fig. 4. Our framework consists of three main parts: Input, Ego-Scene Interactive Modeling, and Task-aware Enhancement. Taking multi-view videos as input, the image backbone first extracts image features. Then, the spatio-temporal flow features are built with

Table 2: Object detection results on nuScenes Caesar et al. (2020) and Bench2Drive Jia et al. (2024). The baseline is SparseBEV.

| Method | Backbone | Input Size | Dataset | mAP↑ | NDS↑ | mATE↓ | mASE↓ | mAOE↓ | mAVE↓ | mAAE↓ |
|---|---|---|---|---|---|---|---|---|---|---|
| StreamPETR Wang et al. (2023) | ResNet50 | 256 × 704 | nuScenes | 0.432 | 0.540 | 0.581 | 0.272 | 0.413 | 0.295 | 0.195 |
| SparseBEV Liu et al. (2023) | ResNet50 | 256 × 704 | nuScenes | 0.445 | 0.553 | 0.605 | 0.278 | 0.389 | 0.253 | 0.189 |
| FlowAD (Ours) | ResNet50 | 256 × 704 | nuScenes | **0.475** | **0.574** | 0.585 | 0.265 | 0.361 | 0.236 | 0.184 |
| BEVDepth Li et al. (2023a) | ResNet101 | 512 × 1408 | nuScenes | 0.412 | 0.535 | 0.565 | 0.266 | 0.358 | 0.331 | 0.190 |
| SparseBEV Liu et al. (2023) | ResNet101 | 512 × 1408 | nuScenes | 0.501 | 0.592 | 0.562 | 0.265 | 0.321 | 0.243 | 0.195 |
| FlowAD (Ours) | ResNet101 | 512 × 1408 | nuScenes | **0.528** | **0.607** | 0.552 | 0.253 | 0.311 | 0.230 | 0.192 |
| BEVFormer Li et al. (2022) | ResNet101 | 900 × 1600 | Bench2Drive | 0.634 | 0.671 | - | - | - | - | - |
| SparseBEV Liu et al. (2023) | ResNet101 | 512 × 1408 | Bench2Drive | 0.641 | 0.689 | 0.386 | 0.165 | 0.124 | 0.189 | - |
| FlowAD (Ours) | ResNet101 | 512 × 1408 | Bench2Drive | **0.673** | **0.723** | 0.346 | 0.150 | 0.117 | 0.184 | - |

the Ego-Scene Interactive Modeling, serving as the input of the Task-aware Enhancement. As the downstream tasks are generally classified into object-level (*e.g.,* object detection) and region-level ones (*e.g.,* VLM to analyse the driving scenario), we design corresponding enhancement strategies.

**Object-level Enhancement.** For object-level tasks that employ $M$ object queries $\mathbf{Q}_{\text{obj}} \in \mathbb{R}^{M \times C}$ for inference (*e.g.,* detection and motion prediction), we enhance the perception capability by applying the spatio-temporal flow feature. As in Fig. 4, object queries are regressed to the sampling points $\mathbf{p} \in \mathbb{R}^{M \times P \times C}$ and projected to multi-view image planes. The flow units covering the sampling points of an object query are exploited to enhance the query embedding with a cross-attention, which injects the spatio-temporal dynamics.

**Region-level Enhancement.** For region-level tasks that perform inference based on the feature of the driving scenario (*e.g.,* constructing the ego query with the observation feature in ego planning, generating scene descriptions in the VLM analysis), we design the region-level enhancement strategy. As shown in Fig. 4, the region features are directly concatenated with the corresponding flow units, with a convolution layer to reduce the feature channels. This helps the model better understand the dynamics of the ego-scene interaction and make more robust decisions.

## 4 EXPERIMENT

### 4.1 EXPERIMENTAL SETUP

**Baselines.** To demonstrate the generality and effectiveness of the proposed FlowAD, we conduct experiments on the tasks of perception, E2E planning, and VLM analysis. **1)** For perception, we apply our framework on SparseBEV Liu et al. (2023), which perceives the scene with sparse object queries. **2)** For E2E planning, SparseDrive Sun et al. (2024) and DiffusionDrive Liao et al. (2025) are adopted as the baselines. **3)** For VLM analysis, we employ Senna Jiang et al. (2024) as the baseline. It should be noted that the official checkpoint is trained on the DriveX dataset which is not released. Thus, we finetune the official Senna on the nuScenes dataset Caesar et al. (2020) to ensure fair comparison. All the inputs and parameters are aligned with the official settings of the baselines.

**Datasets.** We evaluate FlowAD on nuScenes Caesar et al. (2020) and Bench2Drive Jia et al. (2024). **1)** The nuScenes dataset comprises 1,000 driving scenes, with 700 and 150 sequences allocated for training and validation. We perform tasks of perception, end-to-end planning (open-loop) and VLM analysis on this dataset. The metrics are aligned with the baseline methods. **2)** The Bench2Drive dataset is a benchmark to evaluate driving abilities in a closed-loop manner, with 2 million frames collected from CARLA Dosovitskiy et al. (2017). We conduct experiments of perception and end-to-end planning (both open- and closed-loop) on this dataset. Following the official setting, the 950 clips are allocated for training and 50 clips for validation (*i.e.,* perception and open-loop planning). The closed-loop planning evaluation is performed on the official 220 routes. All metrics are aligned with previous works.

**Evaluation for Scene Understanding.** Reviewing commonly used planning metrics (*e.g.,* L2 error and collision rate Caesar et al. (2020)), the capability of scene understanding is not specifically

Table 3: Comparison of 3D occupancy prediction on the Occ3D-nuScenes Tian et al. (2023) dataset. The baseline is SparseOcc Liu et al. (2024).

| Method | Backbone | RayIoU | RayIoU$_{1m,2m,4m}$ | | | FPS |
|---|---|---|---|---|---|---|
| SimpleOcc Gan et al. (2024) | ResNet101 | 22.5 | 17.0 | 22.7 | 27.9 | 9.7 |
| BEVFormer Li et al. (2022) | ResNet101 | 32.4 | 26.1 | 32.9 | 38.0 | 3.0 |
| BEVDet-Occ Huang & Huang (2022) | ResNet50 | 32.6 | 26.6 | 33.1 | 38.2 | 0.8 |
| FB-Occ Li et al. (2023b) | ResNet50 | 33.5 | 26.7 | 34.1 | 39.7 | 10.3 |
| SparseOcc Liu et al. (2024) | ResNet50 | 35.7 | 29.3 | 36.5 | 41.4 | **17.3** |
| FlowAD (Ours) | ResNet50 | **38.4** | **32.2** | **39.3** | **43.8** | 14.6 |

Table 4: Results of perception, motion prediction and planning on the nuScenes Caesar et al. (2020) validation set. The FPS is measured on an RTX3090 GPU. The baselines are SparseDrive Sun et al. (2024) and DiffusionDrive Liao et al. (2025). ‡ denotes using ego status in the planning module.

| Method | Backbone | Detection | | Tracking | | Online Map | Motion Prediction | | Planning | | | FPS ↑ |
|---|---|---|---|---|---|---|---|---|---|---|---|---|
| | | mAP↑ | NDS ↑ | AMOTA↑ | AMOTP↓ | mAP↑ | minADE↓ | minFDE↓ | Avg.L2↓ | Avg.Col↓ | FCP↓ | |
| VAD Jiang et al. (2023) | ResNet50 | - | - | - | - | 0.476 | - | - | 0.72 | 0.21 | 3.07 | 5.2 |
| MomAD Song et al. (2025) | ResNet50 | 0.423 | 0.531 | 0.391 | 1.243 | 0.559 | 0.61 | 0.98 | 0.60 | 0.09 | - | 7.8 |
| DriveWM Wang et al. (2024b) | - | - | - | - | - | - | - | - | 0.80 | 0.26 | - | - |
| LAW Li et al. (2024) | - | - | - | - | - | - | - | - | 0.61 | 0.30 | - | - |
| UncAD Yang et al. (2025) | ResNet50 | - | - | - | - | - | - | - | 0.60 | 0.07 | - | - |
| FocalAD Sun et al. (2025) | ResNet50 | - | - | - | - | - | 0.61 | 0.95 | 0.60 | 0.09 | - | - |
| GenAD Zheng et al. (2024) | ResNet50 | 0.290 | - | - | - | - | - | - | 0.58 | 0.16 | 1.78 | 6.7 |
| ORION (VLM) Fu et al. (2025) | - | - | - | - | - | - | - | - | 0.63 | 0.37 | - | - |
| SparseDrive Sun et al. (2024) | ResNet50 | 0.418 | 0.525 | 0.386 | 1.254 | 0.551 | 0.62 | 0.99 | 0.61 | 0.08 | 2.55 | **9.0** |
| FlowAD$_{SparseDrive}$ (Ours) | ResNet50 | 0.448 | 0.555 | 0.427 | 1.181 | 0.582 | **0.58** | **0.95** | 0.56 | 0.06 | **1.03** | 7.6 |
| DiffusionDrive Liao et al. (2025) | ResNet50 | 0.412 | 0.522 | 0.374 | 1.246 | 0.559 | 0.64 | 1.01 | 0.57 | 0.08 | 1.35 | 6.8 |
| FlowAD$_{DiffusionDrive}$ (Ours) | ResNet50 | **0.477** | **0.567** | **0.466** | **1.141** | **0.605** | 0.61 | 0.97 | **0.54** | **0.04** | 1.05 | 5.4 |
| UniAD Hu et al. (2023) | ResNet101 | 0.380 | 0.498 | 0.359 | 1.320 | - | 0.71 | 1.02 | 0.69 | 0.12 | 2.69 | 1.8 |
| SparseDrive Sun et al. (2024) | ResNet101 | 0.496 | 0.588 | 0.501 | 1.085 | 0.562 | 0.60 | 0.96 | 0.58 | 0.06 | 2.30 | **7.3** |
| FlowAD$_{SparseDrive}$ (Ours) | ResNet101 | **0.523** | **0.605** | **0.518** | **1.040** | **0.595** | **0.56** | **0.93** | **0.52** | 0.05 | **0.91** | 5.4 |
| DriveDreamer‡ Wang et al. (2024a) | - | - | - | - | - | - | - | - | 0.29 | 0.15 | - | - |
| SSR‡ Li & Cui (2024) | ResNet50 | - | - | - | - | - | - | - | 0.39 | 0.06 | 1.91 | **19.6** |
| DriveTransformer‡ Jia et al. (2025) | ResNet50 | - | - | - | - | - | - | - | 0.40 | 0.11 | - | - |
| FlowAD$_{SparseDrive}^{‡}$ (Ours) | ResNet50 | **0.451** | **0.554** | **0.430** | **1.198** | **0.576** | **0.57** | **0.94** | **0.25** | **0.04** | **0.75** | 7.6 |

evaluated, which is critical for robust maneuvering. Then, we introduce a novel Frames before Correct Planning (FCP) metric. As a practised planner can quickly scan the scenario and make reasonable plans with the given command, the frames elapsed until the initiation of a rational action serve as the measurement,

$$\text{FCP} = \frac{1}{N_{cmd}} \sum_{n=1}^{N_{cmd}} \sum_{f=1}^{F_n} (\prod_{h=1}^{f} \mathbb{1}\{|\mathbf{P}_{3s}^h - \mathbf{G}_{3s}^h| \geq 0.5m\}), \tag{8}$$

where $N_{cmd}$ is the number of given commands, $\mathbf{P}_{3s}^h$ and $\mathbf{G}_{3s}^h$ denote the predicted/GT ego trajectories at 3s in the $h$-th frame of the $n$-th video clip ($1 \leq h \leq F_n$). $0.5m$ is the threshold judging if the planned results accord with the command (more thresholds and an average version are provided in the appendix). The FCP metric provides a statistical view to reflect a planner's comprehension of the driving process with ego-scene interaction, which helps to complement the evaluation system.

## 4.2 COMPARISON WITH STATE-OF-THE-ARTS

**Perception of Object Detection.** Tab. 2 compares the results of 3D object detection on the nuScenes and Bench2Drive datasets. Our FlowAD achieves superior performance in both real driving scenarios (nuScenes) and simulated worlds (Bench2Drive), *i.e.,* 52.8%/67.3% mAP and 60.7%/72.3% NDS, respectively. Notably, by applying our framework, the baseline SparseBEV Liu et al. (2023) with the ResNet50 achieves impressive 3.0% mAP and 2.1% NDS gains on the nuScenes dataset. It proves the effectiveness of our ego-scene interactive modeling to improve the perception capability.

**Perception of 3D Occupancy Prediction.** The proposed ego-scene interactive modeling helps to improve the scene comprehension by learning the flow dynamics. We further verify this on vision-centric 3D occupancy prediction, which focuses on partitioning 3D scenes into structured grids. As shown in Tab. 3, our FlowAD brings gains of 2.7% RayIoU to the baseline SparseOcc Liu et al. (2024), proving its effectiveness.

Table 5: Planning results on the Bench2Drive dataset Jia et al. (2024). The baseline method is SparseDrive Sun et al. (2024).

| Method | Backbone | Open-loop Avg. L2↓ | Closed-loop | | | | Closed-loop Ability | | | | | |
|---|---|---|---|---|---|---|---|---|---|---|---|---|
| | | | DS↑ | SR↑ | Effi↑ | Smooth↑ | Merging↑ | Overtaking↑ | Emergency Brake↑ | Give Way↑ | Traffic Sign↑ | Mean↑ |
| UniAD Hu et al. (2023) | ResNet101 | 0.73 | 45.81 | 16.36 | 129.21 | 43.58 | 14.10 | 17.78 | 21.67 | 10.00 | 14.21 | 15.55 |
| VAD Jiang et al. (2023) | ResNet50 | 0.91 | 42.35 | 15.00 | 157.94 | 46.01 | 8.11 | 24.44 | 18.64 | 20.00 | 19.15 | 18.07 |
| SparseDrive Sun et al. (2024) | ResNet50 | 0.83 | 44.54 | 16.71 | 170.21 | 48.63 | 12.50 | 17.50 | 20.00 | 20.00 | 23.03 | 18.60 |
| MomAD Song et al. (2025) | ResNet50 | 0.82 | 47.91 | 18.11 | 174.91 | 51.20 | - | - | - | - | - | - |
| FlowAD (Ours) | ResNet50 | **0.73** | **51.77** | **22.02** | **178.42** | **53.89** | **20.00** | **28.67** | **25.00** | **25.00** | **28.42** | **25.42** |

Table 6: Results of VLM high-level planning on the nuScenes Caesar et al. (2020) validation set. The FPS is measured on an NVIDIA RTX3090 GPU. The baseline is Senna Jiang et al. (2024). * denotes finetuned on nuScenes dataset.

| Method | Planning Accuracy ↑ | Path (F1 Score) | | | Speed (F1 Score) | | | | FPS ↑ |
|---|---|---|---|---|---|---|---|---|---|
| | | Go Straight ↑ | Turn Left ↑ | Turn Right ↑ | Keep ↑ | Accelerate ↑ | Decelerate ↑ | Stop ↑ | |
| Senna Jiang et al. (2024) | 18.67 | 67.81 | 6.99 | 0.00 | 78.11 | 0.00 | 2.12 | 31.71 | 0.43 |
| Senna* Jiang et al. (2024) | 88.54 | 96.85 | 30.53 | 46.94 | 96.50 | 0.00 | 0.00 | 90.82 | **0.43** |
| FlowAD (Ours) | **90.99** | **97.76** | **60.71** | **68.17** | **96.97** | **51.53** | **40.38** | **93.55** | 0.38 |

Table 7: Ablation on the proposed modules in FlowAD. The FPS is measured on the detection baseline SparseBEV Liu et al. (2023). "@3s" denotes the planning metric at 3s.

| # | Ego-guided Scene Partition | | | | Flow Prediction | | Detection | Planning | | VLM Analysis | FPS↑ |
|---|---|---|---|---|---|---|---|---|---|---|---|
| | Start | Multi | Aggre. | Adjust | Spatial | Temporal | mAP↑ | L2@3s↓ | FCP↓ | Planning Acc↑ | |
| ① | | | | | | | 0.445 | 0.96 | 2.55 | 88.54 | **21.7** |
| ② | | | | | ✓ | | 0.454 | 0.93 | 2.23 | 89.13 | 20.9 |
| ③ | | | | | ✓ | ✓ | 0.459 | 0.91 | 1.87 | 89.66 | 20.3 |
| ④ | ✓ | | | | ✓ | ✓ | 0.463 | 0.88 | 1.31 | 90.22 | 20.2 |
| ⑤ | ✓ | ✓ | | | ✓ | ✓ | 0.466 | 0.87 | 1.16 | 90.30 | 19.4 |
| ⑥ | ✓ | ✓ | ✓ | | ✓ | ✓ | 0.471 | 0.86 | 1.13 | 90.68 | 19.1 |
| ⑦ | ✓ | ✓ | ✓ | ✓ | ✓ | ✓ | **0.475** | **0.84** | **1.03** | **90.99** | 18.9 |

**End-to-End Planning.** The open- and closed-loop planning results are listed in Tab. 4 and Tab. 5, respectively. In the open-loop nuScenes dataset, our FlowAD$_{SparseDrive}$ and FlowAD$_{DiffusionDrive}$ achieve lower L2 error and collision rates compared with the baseline SparseDrive Sun et al. (2024) and DiffusionDrive Liao et al. (2025), as well as other SOTAs. This proves the effectiveness of our spatio-temporal flow dynamics to help understand the driving process. Notably, the FCP of our FlowAD$_{SparseDrive}$ is only 0.91 frames, indicating the improved response speed to the given command with our ego-scene interactive modeling. Our planners also excel in the preceding subtasks, which demonstrates the generality. We further evaluate in the closed-loop Bench2Drive dataset, which covers 44 interactive scenes (*e.g.,* cut-ins, overtaking, detours) and 220 routes across diverse weather conditions. Our method impressively improves the driving score (DS) by 7.23% and success rate (SR) by 5.31% over the baseline SparseDrive, as well as the SOTA performance on various metrics. This proves the enhanced planning capability by modeling the ego-scene interactive dynamics in the latent feature space.

**VLM Planning.** Tab. 6 lists the results of VLM high-level planning. Compared with the baseline Senna Jiang et al. (2024) finetuned on nuScenes, our FlowAD improves the accuracy by impressive 2.45%. Notably, in safety-critical steering scenarios (*i.e.,* turning left/right), the F1 scores significantly increase from 30.53%/46.94% to 60.71%/68.17%. This proves the effectiveness of our design to help understand the driving process and make reasonable plans.

## 4.3 COMPONENT-WISE ABLATION

We analyze the influence of each component based on FlowAD of ResNet50, as in Tab. 7. We first ablate the spatial and temporal flow predictions (①~③), which model the core ego-scene interactive dynamics to benefit downstream tasks. Notably, the ego-guided scene partition is not performed, in which flow units are directly divided from multi-view image features (①). By applying the spatial flow prediction, it obtains considerable gains of 0.9% mAP and 0.32 FCP (② *v.s.* ①). The temporal flow prediction further improves the performance with outstanding 45.9% mAP and 1.87 FCP (③). This verifies that perceiving the scene flow helps to understand the driving scenario and enhance the planning capability. We then ablate the influence of the ego-guided scene partition (④~⑦). It

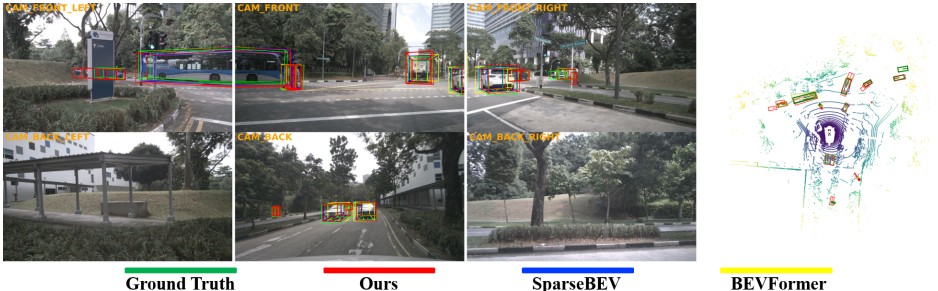

Figure 5: Qualitative results of multi-view object detection on nuScenes Caesar et al. (2020). The baseline is SparseBEV Liu et al. (2023).

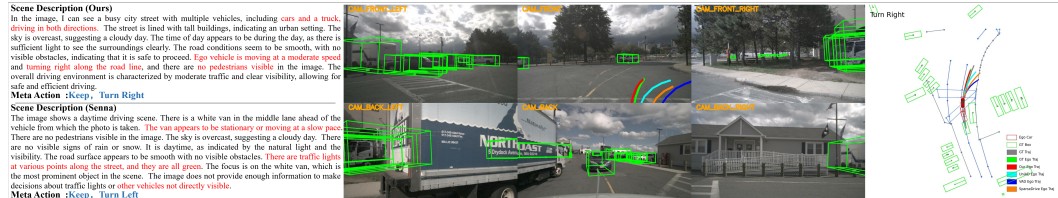

Figure 6: Qualitative results of end-to-end planning and VLM analysis on nuScenes Caesar et al. (2020).

surprises us that the simple starting point of partition significantly improves for 0.4% mAP and 0.56 FCP with only 0.1 FPS cost (④ *v.s.* ③). This proves the necessity of considering the feedback of ego motion, which contributes to superior driving cognition and planning acumen. The multi-level partition, local aggregation and dynamic adjustment also bring considerable gains. The version with all the modules (⑦) has the best performance with 47.5% mAP, 0.14% collision rate at 3s and 90.99% planning accuracy, showing the cooperation of our designs.

## 4.4 VISUALIZATION

We visualize the results of perception, E2E planning and VLM analysis for deeper insights into our designs. As shown in Fig. 5, our detector reveals stronger perception ability compared to the baseline SparseBEV Liu et al. (2023) and BEVFormer Li et al. (2022). Notably, with the learned flow dynamics, our FlowAD is capable of handling partly-occluded objects (front-left camera), proving its superiority. Fig. 6 shows the predicted ego trajectories and high-level VLM planning. Benefited from the enhanced scene understanding, our method could plan a more reasonable ego route under challenging steering scenarios. Besides, the scene description of our method accurately captures surrounding elements and makes reliable planning meta-actions. These further demonstrate the effectiveness of our design to improve diverse tasks by introducing the feedback of the ego motion.

Due to space limitations, we provide additional analyses and experiments in the appendix, covering scene flow, the ego-guided scene partition module, the flow prediction module and the FCP metric. We also demonstrate the effectiveness and robustness of FlowAD and present more visualizations.

## 5 CONCLUSION

We propose FlowAD, a general flow-based framework with ego-scene interactive modeling for autonomous driving. The feedback of ego motion is formulated as the scene flow relative to the ego vehicle. This makes it possible to model the latent interactive dynamics through feature learning with off-the-shelf datasets, instead of simulating. The ego-guided scene partition quantities the scene flow into basic flow units, as well as manifesting the ego motion. The subsequent spatial and temporal flow predictions capture the transition dynamics of spatial displacement and temporal variation based on flow units. Finally, the built spatio-temporal flow dynamics are exploited to benefit object- and region-level tasks. Extensive experiments in both open- and closed-loop evaluations prove the generality and effectiveness of our framework on tasks of perception, end-to-end planning and VLM analysis, as well as the proposed FCP metric.

## ETHICS STATEMENT

This work focuses on ego-scene interactive modeling for autonomous driving. All experiments are conducted on publicly available benchmarks, including NuScenes and Bench2Drive. No human subjects, sensitive personal data, or privacy-related information are involved in this research. The proposed method does not present foreseeable risks of harmful misuse, and the study has no conflicts of interest or commercial sponsorship that could bias the results. We adhered to the ICLR Code of Ethics to ensure fairness, transparency, and academic integrity throughout the research process.

## REPRODUCIBILITY STATEMENT

We have made extensive efforts to ensure the reproducibility of our results. The experimental setup including datasets (NuScenes and Bench2Drive) and evaluation metrics (*e.g.*, L2 error and collision rate), is clearly described in the main text. Comparative baselines (VAD, SparseDrive, *etc*.) and detailed ablation studies are reported to validate our claims. The relevant source code of the core components in this paper has been submitted as anonymous supplementary material. Upon acceptance, we will release all code and related resources through a public code repository on GitHub.

## LLM USAGE STATEMENT

The authors utilized LLMs such as Google's Gemini to refine the language and enhance the readability of this paper. It is important to state that these models were not used for generating ideas or the conceptual framework. All authors have reviewed the final manuscript and take full responsibility for its content and all claims.

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

## A  APPENDIX

The appendix presents additional designing and explaining details of our general flow-based framework for autonomous driving (FlowAD) in the manuscript.

1. **Methodological Elaborations & Discussions**
   - **Motivation of Scene Flow:** We discuss the motivation of the scene flow, which originates from the relative motion between the ego vehicle and the surrounding driving scenario.
   - **Difference with World-Model-Style Approaches:** We elaborate on the methodological differences between our design and other world-model-based methods.

2. **Detailed Analysis of Ego-guided Scene Partition**
   - **Determination of Steering Circle:** We explain the process of determining the steering circle in the Dynamic Adjustment of Partition Size, which decides the adjustment ratios for the ego-left/right scenes.
   - **Different Partition Sizes:** We compare the performance of different partition sizes to manufacture the basic flow units in the proposed Ego-guided Scene Partition.
   - **Influence of Dynamic Adjustment Parameter:** We ablate different coefficients in the dynamic adjustment of partition size to better introduce the ego-motion messages.
   - **Different Ranges of Local Aggregation:** We explore the influence of different ranges in the Local Aggregation to fuse messages for the possible object fragmentations.
   - **Adaptive Aggregation Strategy:** We explore an adaptive aggregation strategy with Deformable Attention, which dynamically fuses local messages.

3. **Detailed Analysis of Flow Prediction Module**
   - **Decoupling of Transition State and Observation Feature:** We explore the influence of decoupling transition state and observation feature in the proposed flow prediction module, which is clearly different from standard world models.

- **Different Levels for Flow Prediction:** We compare features of different levels to perform spatial and temporal flow predictions, which aims to achieve a balance between efficiency and computation cost.
- **Influence of Flow Prediction Loss:** We ablate the self-supervised prediction losses of the spatial and temporal flow prediction modules, which enable the model to forecast the scene flow dynamics.

4. **Details on FCP Metric**

- **Different Thresholds for FCP Metric:** We apply different thresholds for the proposed FCP metric, which reflects the scene understanding of a planner to make quick responses with the given command.
- **Extension of FCP Metric to Closed-loop Scenarios:** We extend the FCP metric by eliminating the requirement of the GT trajectory, which is flexible for both open-loop and closed-loop evaluations.

5. **Effectiveness and Robustness Analysis**

- **Scale of Training Data:** We explore the influence of training with different scales of data, which demonstrates the strong capability of our method to learn from limited samples.
- **Accuracies of Different Scenarios in E2E Planning:** We evaluate the performance of different driving scenarios in E2E planning, which proves the reliability under challenging steering scenarios.
- **Accuracies of Different Scenarios in VLM Analysis:** We detail the accuracy of different driving scenarios in VLM analysis, which demonstrates the influence of training data distribution.
- **Robustness Analysis:** We assess the robustness of our planner by perturbing the input multi-view images, calibration parameters, and sampled timestamps.

6. **Qualitative Visualizations**

- **Visualizations of Various Tasks:** We provide more visualization results for the tasks of perception, end-to-end planning and VLM analysis.
- **Visualizations of Challenging Cases:** We present several visualization cases to illustrate the robustness of our method under complex driving scenarios.
- **Visualization of Activation Maps:** We visualize the activation maps before and after our flow prediction, which illustrates the positive influence of concentrating critical objects.

## B   Methodological Elaborations & Discussions

### B.1   Motivation of Scene Flow

Drawing inspiration from the human perceptual-motor processes, particularly the concept of relative motion Davis & Bobick (1997); Bobick & Davis (2001), we represent the ego-scene interaction as the scene flow relative to the ego-vehicle. As shown in Fig. 7, when the ego vehicle moves forward, the surrounding scene will separate from the center of the front-view camera and move relatively backward along the left and right sides. The flowing driving scenario, including the involved object (*e.g.,* the car and pedestrian), reflects the feedback of the ego motion, hence helping the auto-driving system to understand the driving process by learning the dynamics. We then propose the ego-scene interactive modeling architecture to perceive the optic flow in the latent feature space for anticipatory planning and navigation. The scene is divided into multiple flow units with the guidance of ego motion, which provides the foundation to capture the spatio-temporal transition dynamics, followed by the enhancement strategies to benefit diverse downstream tasks. Notably, local aggregation and multi-level partition strategies are integrated to mitigate the potential object fragmentation and capture scene flows across varied receptive fields. This pipeline enables the modeling of ego-motion feedback using readily available, pre-recorded datasets, obviating the need for complex simulations to generate varied observational outcomes.

### B.2   Difference with World-Model-Style Approaches

As described in the manuscript, we exploit the mechanism of the world model to forecast the transition dynamics in both spatial and temporal dimensions. Here we further clarify the difference

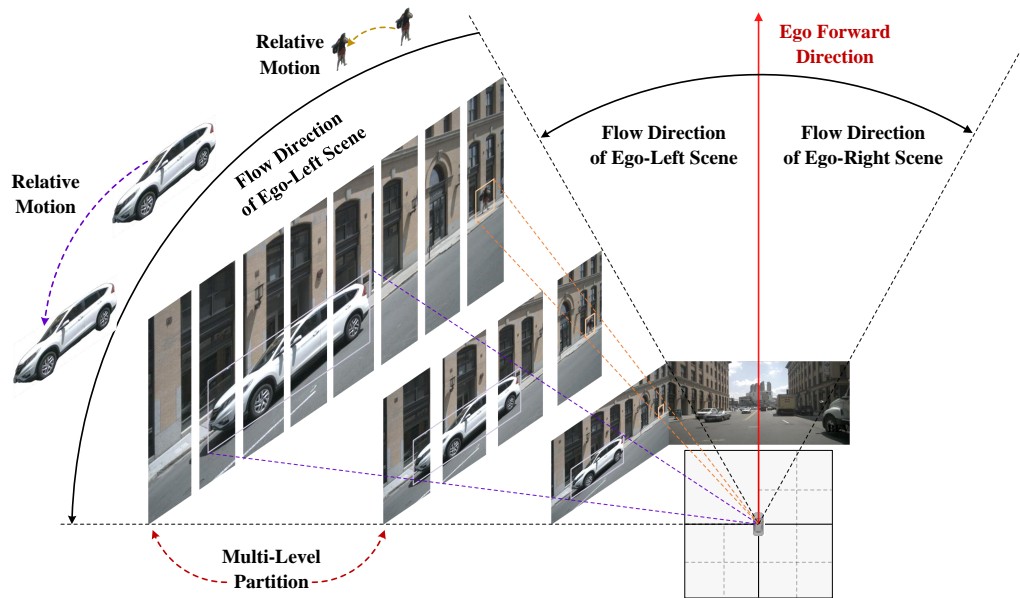

Figure 7: Visualization of the scene flow caused by the relative motion of the ego vehicle.

compared with other world-model-style approaches (*e.g.,* DriveDreamer Wang et al. (2024a) and Drive-WM Wang et al. (2024b)). These planners regard the whole driving environment as a single feature embedding and directly predict the future state driven by the action embedding. Although achieving remarkable driving performance, it neglects the fine-grained environmental messages and the physical influence of ego motion on the camera observations. In contrast, our method divides the driving scenario into multiple parts and extracts the dynamics by flow units. The feedback of ego-motion is explicitly captured by adjusting the start point and ego-left/right sizes of the flow unit partition. These make our ego-scene interactive modeling more sensitive to surrounding dynamics and help the planner understand the driving process and predict robust ego trajectories. In addition, our scene flow performs on both temporal and spatial dimensions, which further benefits the spatial perception of the environment (Tab. 7 of the manuscript). The performance comparison with the mentioned world-model-style methods in Tab. 4 also corroborates these advantages.

## C    DETAILED ANALYSIS OF EGO-GUIDED SCENE PARTITION

### C.1    DETERMINATION OF STEERING CIRCLE

In the Dynamic Adjustment of Partition Size of Sec. 3.1 of the manuscript, we assume the steering trajectory of the ego-vehicle is part of a circle Wang et al. (2005); Park et al. (2015). The radius $r$ is exploited to dynamically adjust the partition sizes of the ego-left/right scenes with the ego width $w_{ego}$, which helps to cognize the feedback of ego motion. Here we detail the computational process. As common sense, three points decide the formulation of a circle. Thus, we exploit the ego posotions $\{(x_t, y_t), (x_{t-1}, y_{t-1}), (x_{t-2}, y_{t-2})\}$ at time $t/t-1/t-2$ to determine the center $(x_c, y_c)$ and radius $r$ of the steering circle. The derivation results are listed as follows, with key items replaced by letters,

$$
\begin{aligned}
&a = 2(x_t - x_{t-1}), \ b = 2(y_t - y_{t-1}), \\
&c = x_t^2 + y_t^2 - x_{t-1}^2 - y_{t-1}^2, \\
&e = 2(x_{t-1} - x_{t-2}), \ f = 2(y_{t-1} - y_{t-2}), \\
&g = x_{t-1}^2 + y_{t-1}^2 - x_{t-2}^2 - y_{t-2}^2, \\
&x_c = \frac{g \times b - c \times f}{e \times b - a \times f}, \ y_c = \frac{a \times g - c \times e}{a \times f - b \times e}, \\
&r = \sqrt{(x_c - x_t)^2 + (y_c - y_t)^2}.
\end{aligned}
\tag{9}
$$

Table 8: Ablation on the base partition size $P$ on the nuScenes Caesar et al. (2020) validation set.

| # | Multi-level Partition Sizes | | | | mAP↑ | NDS↑ | mATE↓ | mASE↓ | mAOE↓ | mAVE↓ | mAAE↓ | FPS↑ |
|---|---|---|---|---|---|---|---|---|---|---|---|---|
| | Stage1 | Stage2 | Stage3 | Stage4 | | | | | | | | |
| ① | 0 | 0 | 0 | 1 | 0.462 | 0.562 | 0.600 | 0.272 | 0.382 | 0.240 | 0.198 | **21.1** |
| ② | 8 | 4 | 2 | 1 | **0.475** | **0.574** | **0.585** | **0.265** | **0.361** | **0.236** | **0.184** | 18.9 |
| ③ | 16 | 8 | 4 | 2 | 0.470 | 0.570 | 0.594 | 0.268 | 0.363 | 0.244 | 0.197 | 19.3 |
| ④ | 24 | 12 | 6 | 3 | 0.459 | 0.562 | 0.601 | 0.271 | 0.380 | 0.243 | 0.199 | 19.8 |

Table 9: Results of perception, motion prediction and planning on the nuScenes Caesar et al. (2020) validation set. The baseline method is SparseDrive Sun et al. (2024). $^*$ denotes using the last FPN feature for multi-level partition and spatial-temporal flow prediction. $^\triangle$ indicates fusing the transition state and forecasted in the flow prediction module.

| Method | Backbone | Detection | | Tracking | | Online Map | Motion Prediction | | Planning | | | FPS ↑ |
|---|---|---|---|---|---|---|---|---|---|---|---|---|
| | | mAP↑ | NDS ↑ | AMOTA↑ | AMOTP↓ | mAP↑ | minADE↓ | minFDE↓ | Avg.L2↓ | Avg.Col↓ | FCP↓ | |
| SparseDrive Sun et al. (2024) | ResNet50 | 0.418 | 0.525 | 0.386 | 1.254 | 0.551 | 0.62 | 0.99 | 0.61 | 0.08 | 2.55 | **9.0** |
| FlowAD (Ours) | ResNet50 | **0.448** | **0.555** | **0.427** | **1.181** | **0.582** | **0.58** | **0.95** | **0.56** | **0.06** | **1.03** | 7.6 |
| FlowAD$^*$ (Ours) | ResNet50 | 0.434 | 0.544 | 0.416 | 1.198 | 0.567 | 0.59 | 0.96 | 0.57 | 0.06 | 1.45 | 8.5 |
| FlowAD$^\triangle$ (Ours) | ResNet50 | 0.429 | 0.533 | 0.401 | 1.207 | 0.559 | 0.60 | 1.01 | 0.59 | 0.07 | 2.03 | 8.5 |

Table 10: Ablation on the power coefficient in the dynamic adjustment of partition size on the nuScenes Caesar et al. (2020) validation set.

| # | Power Coefficient | mAP↑ | NDS↑ | mATE↓ | mASE↓ | mAOE↓ | mAVE↓ | mAAE↓ |
|---|---|---|---|---|---|---|---|---|
| ① | Linear | 0.469 | 0.569 | 0.595 | 0.271 | 0.363 | 0.240 | 0.190 |
| ② | Quadratic | **0.475** | **0.574** | **0.585** | 0.265 | 0.361 | **0.236** | **0.184** |
| ③ | Cubic | 0.474 | 0.572 | 0.589 | **0.256** | **0.354** | 0.237 | 0.187 |

Then the radius $r$ of the steering circle is exploited to compute the dynamic adjust ratios for the scenes in ego-left/right. This makes the modeling way conform to the kinematic characteristics of ego-scene interaction, which benefits the understanding of the driving process.

## C.2 DIFFERENT PARTITION SIZES

Following the design of the FPN network Ren et al. (2015), we propose a multi-level partition to model the flow dynamics of different receptive fields. Here we explore the influence of different partition sizes as in Tab. 8. It shows that applying the multi-level partition brings the performance gains of considerable 1.3% mAP and 1.2% NDS with patch sizes of $\{8, 4, 2, 1\}$ for the four-level image features (② *v.s.* ①). The effectiveness of our design is proven, which helps to understand the ego-scene interaction with flow units of different sizes. It should be noticed that increasing the partition size would instead degrade the performance (④③ *v.s.* ②). The reason lies in the oversized flow unit, which makes it hard to quantify the scene flow dynamics. This also proves the necessity of our partition strategy to construct basic items for dividing and modeling the scene flow.

Considering the multi-level flow prediction is the primary bottleneck causing the FPS degradation, to optimize this trade-off, we perform the ego-scene interactive modeling only on the last stage of multi-view features. The result is listed in Tab. 9. It shows that the lightweight FlowAD maintains its leading performance while incurring only a minimal drop of 0.5 FPS, achieving a balance between effectiveness and efficiency.

## C.3 INFLUENCE OF DYNAMIC ADJUSTMENT PARAMETER OF PARTITION SIZE

In Sec. 3.1, we choose the quadratic power for the coefficients of ego-left/right partition sizes. Here we further explore the influence by ablating the power, as in Tab. 10. The results show that linear power is not sufficient to reflect the different motion speeds of ego-left/right, which is inferior to the one with quadratic power. Notably, the cubic power is overlarge to balance the partition sizes of flow units on both sides. The orientation-relevant metrics are further improved (e.g., mASE and mAOE), while the overall mAP/NDS is inferior to the quadratic one.

Table 11: Ablation on the range of local aggregation on the nuScenes Caesar et al. (2020) validation set.

| # | Range of Local Aggregation | mAP↑ | NDS↑ | mATE↓ | mASE↓ | mAOE↓ | mAVE↓ | mAAE↓ | FPS↑ |
|---|---|---|---|---|---|---|---|---|---|
| ① | 1 | 0.460 | 0.563 | 0.605 | 0.270 | 0.365 | 0.239 | 0.200 | 19.2 |
| ② | 3 | **0.475** | 0.574 | 0.585 | **0.265** | 0.361 | **0.236** | **0.184** | **18.9** |
| ③ | 5 | 0.471 | **0.575** | **0.574** | 0.269 | **0.350** | 0.238 | 0.196 | 18.8 |
| ④ | 7 | 0.464 | 0.567 | 0.588 | 0.271 | 0.366 | 0.242 | 0.194 | 18.6 |

Table 12: Ablation of aggregation strategies on the nuScenes Caesar et al. (2020) validation set.

| Method | Aggregation | mAP↑ | NDS↑ | mATE↓ | mASE↓ | mAOE↓ | mAVE↓ | mAAE↓ | FPS↑ |
|---|---|---|---|---|---|---|---|---|---|
| SparseBEV Liu et al. (2023) | | 0.445 | 0.553 | 0.605 | 0.278 | 0.389 | 0.253 | 0.189 | 21.7 |
| FlowAD (Ours) | Local | 0.475 | 0.574 | 0.585 | **0.265** | **0.361** | **0.236** | **0.184** | **18.9** |
| FlowAD (Ours) | Deformable | **0.480** | **0.580** | **0.582** | 0.273 | 0.367 | 0.241 | 0.185 | 16.7 |

## C.4 DIFFERENT RANGES OF LOCAL AGGREGATION

As mentioned in Sec. 3.1 of the manuscript, the proposed local aggregation fuses adjacent flow units to handle possible object fragmentations. We then ablate different ranges to explore the influence, as shown in Tab. 11. Compared with the version without local aggregation, fusion with the range of 3 brings considerable gains of 1.5% mAP (② *v.s.* ①), proving its effectiveness for environmental modeling. Notably, it does not bring further improvements when further increasing the fusion range (④③ *v.s.* ②). The underlying reason is that the overbroad fusion would weaken the unique distribution of each flow unit and distract the learning of scene flow dynamics. We adopt the fusion range of 3 as the default setting in our implementation.

## C.5 ADAPTIVE AGGREGATION STRATEGY

We explore the adaptive aggregation by applying the Deformable Attention Xia et al. (2022), which dynamically aggregates the surrounding environmental information for each partitioned patch. Notably, the initial sampling locations are uniformly distributed in local 3 flow units, which are then dynamically adjusted by the predicted offsets. Considering the computation costs caused by the high resolution of features in earlier FPN stages, the aggregation is only performed on the last two stages. The result is listed in Tab. 12. It shows that the mAP/NDS further improves, while the FPS degrades for 2.2 FPS. We will explore more efficient dynamic aggregation strategy in future work.

## D DETAILED ANALYSIS OF FLOW PREDICTION MODULE

### D.1 DECOUPLING OF TRANSITION STATE AND OBSERVATION FEATURE

Different from the standard world models Hafner et al. (2020); Wang et al. (2024b) that regard the constantly-updated transition state as the predicted environmental feature, our flow prediction module decouples the two concepts to unleash the modeling potential. Specifically, the transition state modeled with GRU (same as the standard world model) is exploited to backward update the observation feature with cross-attention (Fig. 3 of the manuscript). The ablation in Tab. 9 further proves that the decoupling helps to better extract environmental information and benefit planning.

### D.2 DIFFERENT LEVELS FOR FLOW PREDICTION

The multi-level partition in Sec. 3.1 of the manuscript is performed to capture the flow dynamics of different receptive fields. We further explore the influence by ablating the feature levels for flow prediction, as shown in Tab. 13. With the spatio-temporal flow prediction on the first level feature, the mAP/NDS metric improves by considerable 1.5%/0.7% (② *v.s.* ①), respectively. It proves the effectiveness of our design to help object perception. An interesting observation is that the performance of the version with flow prediction on the fourth level feature (③) is better than the one on the first level feature (①). This reveals that the high-level feature with more semantic messages brings more benefits to environmental modeling compared with the low-level ones with more tex-

Table 13: Ablation on the levels for flow prediction on nuScenes Caesar et al. (2020) validation set.

| # | Levels for Flow Prediction Stage1 | Stage2 | Stage3 | Stage4 | mAP↑ | NDS↑ | mATE↓ | mASE↓ | mAOE↓ | mAVE↓ | mAAE↓ | FPS↑ |
|---|---|---|---|---|---|---|---|---|---|---|---|---|
| ① | | | | | 0.445 | 0.553 | 0.605 | 0.278 | 0.389 | 0.253 | 0.189 | **21.7** |
| ② | ✓ | | | | 0.460 | 0.560 | 0.599 | 0.272 | 0.381 | 0.247 | 0.190 | 19.8 |
| ③ | | | | ✓ | 0.466 | 0.564 | 0.594 | 0.269 | 0.378 | 0.244 | 0.192 | 20.1 |
| ④ | | | ✓ | ✓ | 0.475 | 0.574 | 0.585 | 0.265 | 0.361 | 0.236 | 0.184 | 18.9 |
| ⑤ | | ✓ | ✓ | ✓ | 0.479 | 0.580 | 0.581 | 0.266 | 0.357 | **0.232** | **0.183** | 17.2 |
| ⑥ | ✓ | ✓ | ✓ | ✓ | **0.481** | **0.583** | **0.578** | **0.263** | **0.356** | 0.233 | 0.184 | 15.1 |

Table 14: Ablation on the flow prediction losses on the nuScenes Caesar et al. (2020) validation set.

| # | Flow Prediction Loss Spatial | Temporal | mAP↑ | NDS↑ | mATE↓ | mASE↓ | mAOE↓ | mAVE↓ | mAAE↓ |
|---|---|---|---|---|---|---|---|---|---|
| ① | | | 0.444 | 0.548 | 0.619 | 0.280 | 0.392 | 0.256 | 0.192 |
| ② | ✓ | | 0.459 | 0.562 | 0.596 | 0.273 | 0.379 | 0.244 | 0.187 |
| ③ | | ✓ | 0.456 | 0.559 | 0.601 | 0.276 | 0.381 | 0.246 | 0.189 |
| ④ | ✓ | ✓ | **0.475** | **0.574** | **0.585** | **0.265** | **0.361** | **0.236** | **0.184** |

Table 15: Influence of the flow prediction losses on the nuScenes Caesar et al. (2020) validation set. The default train epoch is 24, following the baseline SparseBEV Liu et al. (2023).

| # | Method | Train Epoch | Flow Prediction Loss Spatial | Temporal | mAP↑ | NDS↑ | mATE↓ | mASE↓ | mAOE↓ | mAVE↓ | mAAE↓ |
|---|---|---|---|---|---|---|---|---|---|---|---|
| ① | SparseBEV Liu et al. (2023) | 18 | | | 0.437 | 0.541 | 0.625 | 0.283 | 0.401 | 0.255 | 0.191 |
| ② | SparseBEV Liu et al. (2023) | 24 | | | 0.445 | 0.553 | 0.605 | 0.278 | 0.389 | 0.253 | 0.189 |
| ③ | SparseBEV Liu et al. (2023) | 30 | | | 0.448 | 0.554 | 0.599 | 0.273 | 0.389 | 0.256 | 0.193 |
| ④ | FlowAD (Ours) | 18 | | | 0.393 | 0.499 | 0.671 | 0.294 | 0.404 | 0.275 | 0.198 |
| ⑤ | FlowAD (Ours) | 24 | | | 0.444 | 0.548 | 0.619 | 0.280 | 0.392 | 0.256 | 0.192 |
| ⑥ | FlowAD (Ours) | 30 | | | 0.468 | 0.570 | 0.598 | 0.269 | 0.390 | 0.245 | 0.189 |
| ⑦ | FlowAD (Ours) | 18 | ✓ | ✓ | 0.454 | 0.557 | 0.611 | 0.270 | 0.385 | 0.244 | 0.187 |
| ⑧ | FlowAD (Ours) | 24 | ✓ | ✓ | 0.475 | 0.574 | 0.585 | 0.265 | 0.361 | 0.236 | 0.184 |
| ⑨ | FlowAD (Ours) | 30 | ✓ | ✓ | **0.483** | **0.580** | **0.578** | **0.263** | **0.355** | **0.233** | **0.183** |

tural information. The version that performs flow prediction on all-level features achieves the best performance with outstanding 48.1% mAP and 58.3% NDS (⑥), yet causes more computation costs (*i.e.*, 21.7 FPS of ① → 15.1 FPS). To balance the performance and efficiency, we only perform flow prediction on the third- and fourth-level features (④) in our default implementation.

## D.3 INFLUENCE OF FLOW PREDICTION LOSS

Following the loss design in world models Hafner et al. (2020; 2021), we respectively map each predicted/GT flow unit to latent states and minimize their KL divergence. This endows the model with the capability of forecasting scene flow dynamics, hence benefiting the understanding of the feedback of ego motion to the driving scenario. We then ablate the spatial and temporal flow prediction losses to explore the influence. As shown in Tab. 14, the performance of the version removing all flow prediction losses is even inferior to the baseline SparseBEV Liu et al. (2023) (①). This reveals the necessity of explicit supervision to learn effective environmental messages. When applying the spatial or temporal flow prediction losses, the performance improves for 1.5%/1.2% mAP respectively (②③ *v.s.* ①), proving the effectiveness. Notably, the spatial supervision brings more performance gains than the temporal one (③ *v.s.* ②). It demonstrates that spatial knowledge is more helpful for scene understanding. Applying both losses contributes to the best version with 47.5% mAP and 57.4% NDS, showing the cooperation of the spatio-temporal modeling.

We further explore the influence of flow prediction losses by ablating them, *i.e.,* without the explicit supervision of the predicted transition states. As shown in Tab. 15, our model without world model loss benefits more from the training epochs compared with the baseline, which achieves superior performance at the training epoch 30. In contrast, the one with explicit supervision reaches convergence much faster. These reveal that the world model loss helps the model to learn realistic future driving environment, as well as the optimization potential of our design.

Table 16: Planning results in terms of L2 error, collision rate, and the proposed FCP metrics with different thresholds on the nuScenes Caesar et al. (2020) validation set. The baseline method is SparseDrive Sun et al. (2024). $\diamond$ denotes the extended FCP metric.

| Method | Backbone | L2 Error (m) ↓ | | | | Collision Rate (%) ↓ | | | | FCP (frame) ↓ | | | | FCP$^\diamond$ (frame) ↓ | | | | FPS ↑ |
|---|---|---|---|---|---|---|---|---|---|---|---|---|---|---|---|---|---|---|
| | | 1s | 2s | 3s | Avg. | 1s | 2s | 3s | Avg. | 0.25m | 0.50m | 0.75m | Avg. | 1 frame | 2 frame | 3 frame | Avg. | |
| UniAD Hu et al. (2023) | ResNet101 | 0.44 | 0.67 | 0.96 | 0.69 | 0.04 | 0.08 | 0.23 | 0.12 | 5.41 | 2.69 | 0.93 | 3.01 | 0.73 | 2.02 | 3.43 | 2.06 | 1.8 |
| VAD Jiang et al. (2023) | ResNet50 | 0.41 | 0.70 | 1.05 | 0.72 | 0.07 | 0.17 | 0.41 | 0.22 | 6.44 | 3.07 | 1.42 | 3.64 | 0.87 | 2.39 | 4.12 | 2.46 | 5.2 |
| MomAD Song et al. (2025) | ResNet50 | 0.31 | 0.57 | 0.91 | 0.60 | 0.01 | 0.05 | 0.22 | 0.09 | - | - | - | - | - | - | - | - | 7.8 |
| SparseDrive Sun et al. (2024) | ResNet50 | 0.29 | 0.58 | 0.96 | 0.61 | 0.01 | 0.05 | 0.18 | 0.08 | 4.87 | 2.55 | 1.09 | 2.84 | 0.54 | 1.88 | 3.02 | 1.48 | **9.0** |
| FlowAD (Ours) | ResNet50 | **0.28** | **0.55** | **0.84** | **0.56** | **0.01** | **0.03** | **0.15** | **0.06** | **2.12** | **1.03** | **0.48** | **1.21** | **0.37** | **0.90** | **1.79** | **1.02** | 7.6 |

# E  DETAILS ON FCP METRIC

## E.1  DIFFERENT THRESHOLDS FOR FCP METRIC

Having observed the capability of the scene understanding within an auto-driving system is not specifically evaluated in current metrics (*e.g.,* L2 error and collision rate Caesar et al. (2020)), we propose the FCP metric to provide a statistical perspective. The frames elapsed until the initiation of a rational action are exploited as the measurement, with the threshold of $0.5m$ to judge if the planned results accord with the command. Here we ablate more thresholds for deeper exploration, as listed in Tab. 16. The results show that, with a stricter threshold of $0.25m$, the FCP of our method dramatically increases from 1.03 frames ($0.5m$) to 2.12 frames. It indicates the slowed response speed of our method under more rigorous measurement standards. Even though, our FlowAD surpasses other SOTA planners with a large performance margin, *e.g.,* our 2.12 frames *v.s.* 4.87 frames of the baseline SparseDrive Sun et al. (2024), demonstrating the superiority. The less restrictive threshold of $0.75m$ leads to better performance of the FCP metric. The average version of the three thresholds reflects a general comprehension of the driving scenario for a planner, in which our FlowAD still performs best with outstanding 1.21 FCP. This proves the effectiveness of our method to help understand the driving process and make quick responses to the given command.

## E.2  EXTENSION OF FCP METRIC TO CLOSED-LOOP SCENARIOS

We propose the FCP metric to evaluate a planner's reaction speed to the given command by computing the errors compared with the GT trajectory. While reliance on Ground Truth (GT) is standard for open-loop metrics (e.g., L2 Error), it inherently limits the assessment of intrinsic correctness and the multi-modal nature of human-like planning. To overcome this limitation and facilitate a more precise evaluation of reaction speed, we extend the FCP metric to operate beyond simple GT matching. Specifically, a "successful reaction" is defined based on the trajectory's compliance with the high-level command rather than its proximity to the GT. For a given command (applicable in both open- and closed-loop settings), the model is considered responsive if its predicted trajectory at the 3s horizon aligns with the command for a predefined sequence of $q$ frames. For instance, regarding a "Turn Right" command, adopting the threshold from UniAD Hu et al. (2023), a planner is deemed successful if the lateral displacement of the ego trajectory exceeds 2.0m for $q$ consecutive frames. The extended metric is formulated as,

$$\text{FCP}^\diamond = \frac{1}{N_{cmd}} \sum_{n=1}^{N_{cmd}} \max(0, \sum_{f=1}^{F_n} (\prod_{h=1}^{f} \mathbb{1}\{\mathbf{P}_{3s}^h \neq command\}) - q), \qquad (10)$$

We then perform comparison using the extended FCP$^\diamond$ metric, as shown in Tab. 16. The results indicate that our method consistently maintains its superiority over competing approaches across varying thresholds of $q$, validating the robustness of our planner. Moreover, by eliminating GT dependence and accommodating diverse decision-making behaviors, this refined metric offers the flexibility to be deployed in both open-loop and closed-loop settings. We hope FCP$^\diamond$ serves as a standard, robust benchmark for a more holistic assessment of autonomous driving capabilities.

Table 17: Ablation on the scale of training data on the nuScenes Caesar et al. (2020) validation set.

| # | Scale of Training Data | mAP↑ | NDS↑ | mATE↓ | mASE↓ | mAOE↓ | mAVE↓ | mAAE↓ |
|---|---|---|---|---|---|---|---|---|
| ① | 25% | 0.391 | 0.497 | 0.691 | 0.275 | 0.519 | 0.297 | 0.200 |
| ② | 50% | 0.448 | 0.554 | 0.593 | 0.269 | 0.389 | 0.257 | 0.194 |
| ③ | 75% | 0.466 | 0.569 | 0.587 | 0.266 | 0.367 | 0.244 | 0.188 |
| ④ | 100% | **0.475** | **0.574** | **0.585** | **0.265** | **0.361** | **0.236** | **0.184** |

Table 18: Performance comparison under different driving scenarios on the nuScenes Caesar et al. (2020) validation set.

| Method | Backbone | Perf. *go straight* ↓ (5309 samples) | | Perf. *turn left* ↓ (301 samples) | | Perf. *turn right* ↓ (409 samples) | | Perf. *Overall* ↓ (6019 samples) | |
|---|---|---|---|---|---|---|---|---|---|
| | | Avg. L2 (m) | Avg. Col. (%) | Avg. L2 (m) | Avg. Col. (%) | Avg. L2 (m) | Avg. Col. (%) | Avg. L2 (m) | Avg. Col. (%) |
| SparseDrive Sun et al. (2024) | ResNet50 | 0.58 | 0.08 | 0.83 | 0.29 | 0.86 | **0.08** | 0.61 | 0.09 |
| FlowAD$_{SparseDrive}$ | ResNet50 | **0.53** | **0.06** | **0.77** | **0.08** | **0.69** | 0.09 | **0.56** | **0.06** |
| SparseDrive Sun et al. (2024) | ResNet101 | 0.56 | 0.06 | 0.89 | 0.13 | 0.81 | 0.10 | 0.58 | 0.06 |
| FlowAD$_{SparseDrive}$ | ResNet101 | **0.50** | **0.05** | **0.74** | **0.08** | **0.65** | **0.08** | **0.52** | **0.05** |
| DiffusionDrive Liao et al. (2025) | ResNet50 | 0.55 | 0.07 | 0.64 | 0.12 | 0.59 | 0.10 | 0.57 | 0.08 |
| FlowAD$_{DiffusionDrive}$ | ResNet50 | **0.54** | **0.03** | **0.62** | **0.02** | **0.54** | **0.07** | **0.54** | **0.06** |

# F EFFECTIVENESS AND ROBUSTNESS ANALYSIS

## F.1 SCALE OF TRAINING DATA

The quality of ego-scene interactive modeling is deeply influenced by the training data volume. We then explore the influence by training our FlowAD with different scales of data and the results are presented in Tab. 17. The default setting for the scale of training data is noted as "100%". It shows that as the scale of training data reduces (*i.e.,* "75%", "50%" and "25%"), the overall performance gradually decreases (*e.g.,* $47.5\% \rightarrow 46.6\% \rightarrow 44.8\% \rightarrow 39.1\%$ of mAP on the nuScenes dataset), demonstrating that more data helps improve the model capacity of understanding the driving process of ego-scene interaction. Notably, even with only 50% of the training data, our FlowAD has achieved comparable performance with the baseline SparseBEV Liu et al. (2023), proving the effectiveness of our design.

## F.2 ACCURACIES OF DIFFERENT SCENARIOS IN E2E PLANNING

We evaluate the benefit of our ego-scene interactive modeling under different driving scenes, as shown in Tab. 18. According to the given driving command (*i.e., go straight*, *turn left* and *turn right*), we divide the 6,019 validation samples in nuScenes Caesar et al. (2020) into three parts, which contain 5,309, 301 and 409 ones, respectively. As expected, all methods perform better under *go straight* scenes than the steering scenes. When applying the proposed ego-scene interactive modeling, our planners achieve considerable gains in steering scenes, proving its robustness. Furthermore, the ability metrics on more challenging Bench2Drive (Tab. 5 of the manuscript) also confirm the stability of our approach in diverse and extreme maneuver scenarios.

## F.3 ACCURACIES OF DIFFERENT SCENARIOS IN VLM ANALYSIS

The ego-scene interactive modeling is proposed to enhance the comprehension of the driving process and improve the planning capability. We demonstrate the superiority of our proposed architecture by evaluating the accuracies of different driving scenarios in Tab. 19. It shows that our FlowAD achieves consistent performance gains compared with the baseline Senna Jiang et al. (2024). Notably, the nuScenes dataset Caesar et al. (2020) is dominated by the straight scenes (*i.e., 4724* samples of "Keep/Accelerate/Decelerate/Stop Straight" to all 5119 samples). The performance of steering scenarios is less reflected in the overall planning accuracy metric, yet is critical for safe maneuvering. In such "Left/Right Turn" cases, our method still demonstrates superior accuracy, proving the effectiveness of the scene flow modeling to benefit scene understanding and planning.

Table 19: Accuracy of high-level planning on the nuScenes Caesar et al. (2020) validation set. The baseline is Senna Jiang et al. (2024). * denotes finetuned on the nuScenes dataset.

| Method | Planning Accuracy (5119 samples) | Keep Right Turn (187 samples) | Keep Left Turn (139 samples) | Keep Straight (3704 samples) | Accelerate Right Turn (3 samples) | Accelerate Left Turn (4 samples) | Accelerate Straight (62 samples) | Decelerate Straight (45 samples) | Stop Straight (975 samples) | FPS |
|---|---|---|---|---|---|---|---|---|---|---|
| Senna Jiang et al. (2024) | 18.67 | 0.00 | 18.03 | 37.53 | 0.00 | 0.00 | 0.00 | 2.22 | 26.56 | 0.43 |
| Senna* Jiang et al. (2024) | 88.54 | 35.98 | 28.78 | 92.28 | 0.00 | 0.00 | 0.00 | 6.67 | 88.13 | 0.43 |
| FlowAD (Ours) | **90.99** | **69.52** | **69.06** | **94.71** | **66.67** | **25.00** | **43.55** | 17.78 | **90.87** | 0.39 |

Table 20: Robustness analysis on the nuScenes Caesar et al. (2020) validation set. The baseline method is SparseDrive Sun et al. (2024) with ResNet101.

| Method | GaussianBlur Noise | Calibration Error | Timestamp Drift | Detection mAP↑ | NDS↑ | Tracking AMOTA↑ | AMOTP↓ | Online Map mAP↑ | Motion Prediction minADE↓ | minFDE↓ | Planning Avg.L2↓ | Avg.Col↓ | FCP↓ |
|---|---|---|---|---|---|---|---|---|---|---|---|---|---|
| SparseDrive Sun et al. (2024) | | | | 0.496 | 0.588 | 0.501 | 1.085 | 0.562 | 0.60 | 0.96 | 0.58 | 0.06 | 2.30 |
| FlowAD (Ours) | | | | **0.523** | **0.605** | **0.518** | **1.040** | **0.595** | **0.56** | **0.93** | **0.52** | **0.05** | **0.91** |
| SparseDrive Sun et al. (2024) | ✓ | | | 0.472 | 0.555 | 0.487 | 1.159 | 0.533 | 0.63 | 1.03 | 0.60 | 0.07 | 2.34 |
| FlowAD (Ours) | ✓ | | | **0.517** | **0.596** | **0.514** | **1.053** | **0.583** | **0.57** | **0.96** | **0.53** | **0.05** | **0.96** |
| SparseDrive Sun et al. (2024) | | ✓ | | 0.438 | 0.540 | 0.466 | 1.211 | 0.509 | 0.64 | 1.07 | 0.65 | 0.08 | 3.22 |
| FlowAD (Ours) | | ✓ | | **0.486** | **0.570** | **0.498** | **1.103** | **0.543** | **0.58** | **0.99** | **0.56** | **0.06** | **1.55** |
| SparseDrive Sun et al. (2024) | | | ✓ | 0.470 | 0.565 | 0.483 | 1.107 | 0.528 | 0.61 | 1.00 | 0.60 | 0.07 | 3.93 |
| FlowAD (Ours) | | | ✓ | **0.505** | **0.590** | **0.508** | **1.056** | **0.569** | **0.56** | **0.95** | **0.53** | **0.05** | **1.98** |
| SparseDrive Sun et al. (2024) | ✓ | ✓ | ✓ | 0.419 | 0.531 | 0.457 | 1.244 | 0.485 | 0.65 | 1.10 | 0.66 | 0.08 | 4.20 |
| FlowAD (Ours) | ✓ | ✓ | ✓ | **0.465** | **0.558** | **0.476** | **1.119** | **0.520** | **0.60** | **1.02** | **0.57** | **0.06** | **2.17** |

## F.4 ROBUSTNESS ANALYSIS

Here we evaluate the robustness of our planner by (1) injecting Gaussian noise into the observed images, (2) perturbing camera intrinsics/extrinsics and ego-pose parameters by multiplying a random coefficient with the range of $0.95 \sim 1.05$, and (3) introducing temporal drift by randomly sampling the input frame from $t - 2$ to $t$. The results in Tab. 20 show that our method consistently demonstrates stronger robustness compared with the baseline SparseDrive Sun et al. (2024).

## G QUALITATIVE VISUALIZATIONS

### G.1 VISUALIZATIONS OF VARIOUS TASKS

We visualize the results of perception, E2E planning and VLM analysis for deeper insights into our designs, which demonstrate the generality and effectiveness of our designs. As depicted in Fig. 8, our detector reveals a stronger perception ability of the surrounding environment and localizes the objects, compared with the baseline SparseBEV Liu et al. (2023) and BEVFormer Li et al. (2022). It should be noticed that our FlowAD is capable of exploiting the learned flow dynamics to capture hard cases, *e.g.,* part occlusion, small and dense objects. It proves the superiority of our design to enhance the perception ability. Fig. 9 compares the planned ego trajectories and high-level VLM planning with recent SOTA planners. With the proposed ego-scene interactive modeling, our FlowAD could quickly understand the driving scenario and plan a more reasonable ego route under various scenarios. Besides, the scene descriptions of our method accurately capture surrounding elements and make reliable planning meta-actions. We also visualize the planned routes in the closed-loop Bench2Drive dataset Jia et al. (2024), which performs real-time interactions between the ego vehicle and driving scene with the CARLA simulator Dosovitskiy et al. (2017). The Fig. 10 shows that our FlowAD is able to notice the critical driving elements (*e.g.,* pedestrians, cars, and traffic lights) and drive the vehicle in a safe manner. These further prove the effectiveness of our design to improve diverse tasks by learning the feedback of the ego motion in the latent space.

### G.2 VISUALIZATIONS OF CHALLENGING CASES

**Non-ego-centric Driving Scenario.** FlowAD's scene flow modeling captures object dynamics beyond static background through local aggregation that handles object fragmentation. Therefore, our approach essentially models the interactive dynamics between the vehicle, the static environment, and moving objects. This enhances the model's understanding of autonomous driving and its capacity to capture scene dynamics. Consequently, our method can handle non-ego-centric dynamics, such as the "cut-in" scenario. We further demonstrate this by the case in Fig. 11. As a leading

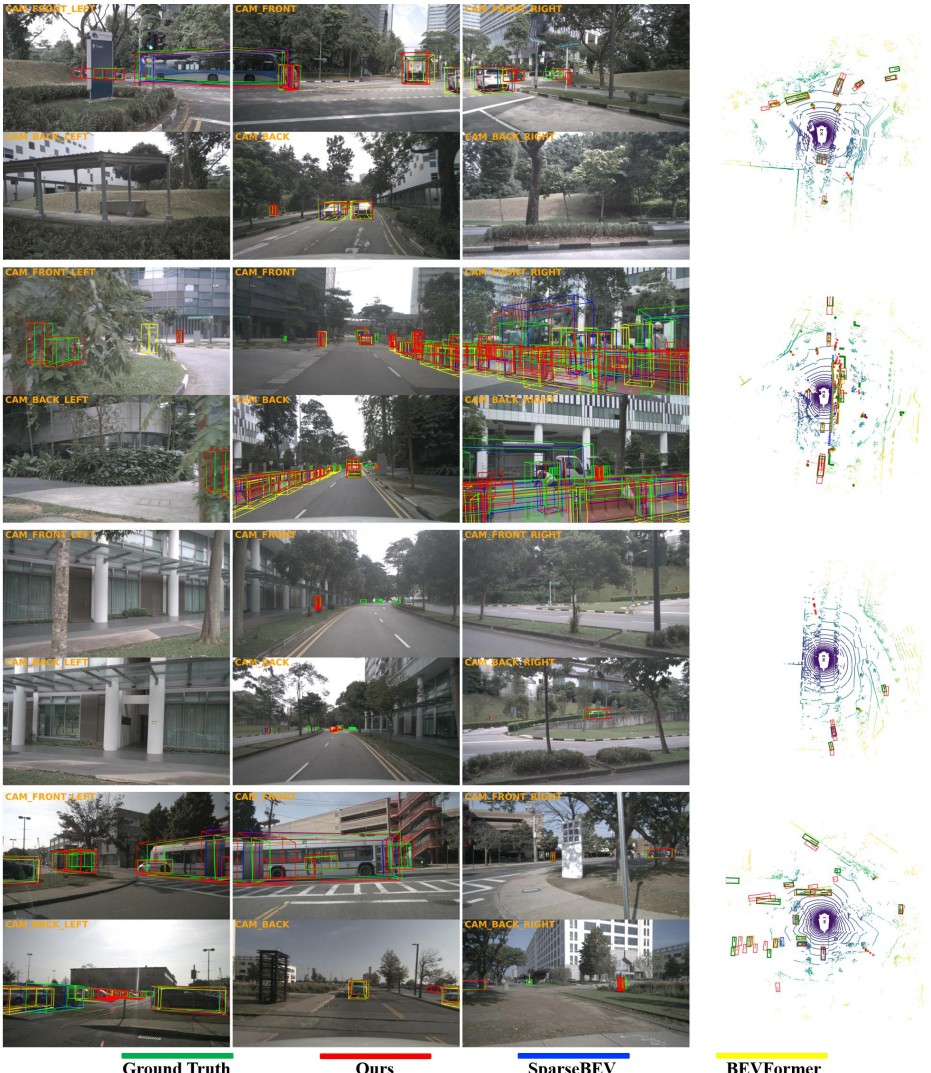

Figure 8: Qualitative results of multi-view object detection on nuScenes Caesar et al. (2020). Results show the strong capability of our method to handle occluded objects with scene flow dynamics.

vehicle performs a cut-in, our planner has already inferred its motion state from historical observations and produces a salient response, corroborating the above hypothesis. Besides, the superiority of the ability metrics in Bench2Drive Jia et al. (2024) (e.g., Merging and Emergency Brake that include many cutting-in scenarios, Tab. 5 of the manuscript) substantiates our method's advantage in modeling vehicles that are not centered on ego-motion.

**Extreme Dynamic Driving Scenario.** The model's stability is further evaluated under challenging longitudinal and topographical conditions, including rapid acceleration, deceleration, and traversing slopes. As shown in Fig. 12, the proposed FlowAD exhibits superior robustness in these hard scenarios. Fig. 13 demonstrates the superior steering capability of our planner with the proposed ego-scene interactive modeling. These qualitative observations align with the quantitative ability metrics reported on Bench2Drive Jia et al. (2024) (Tab. 5 of the manuscript), which validate the system's stability under diverse and extreme dynamic maneuvers.

## G.3 VISUALIZATION OF ACTIVATION MAPS

With the aim of better illustrating the mechanism of the proposed flow prediction, we visualize the activation maps of multi-view image features before and after our design using GradCAM, as in

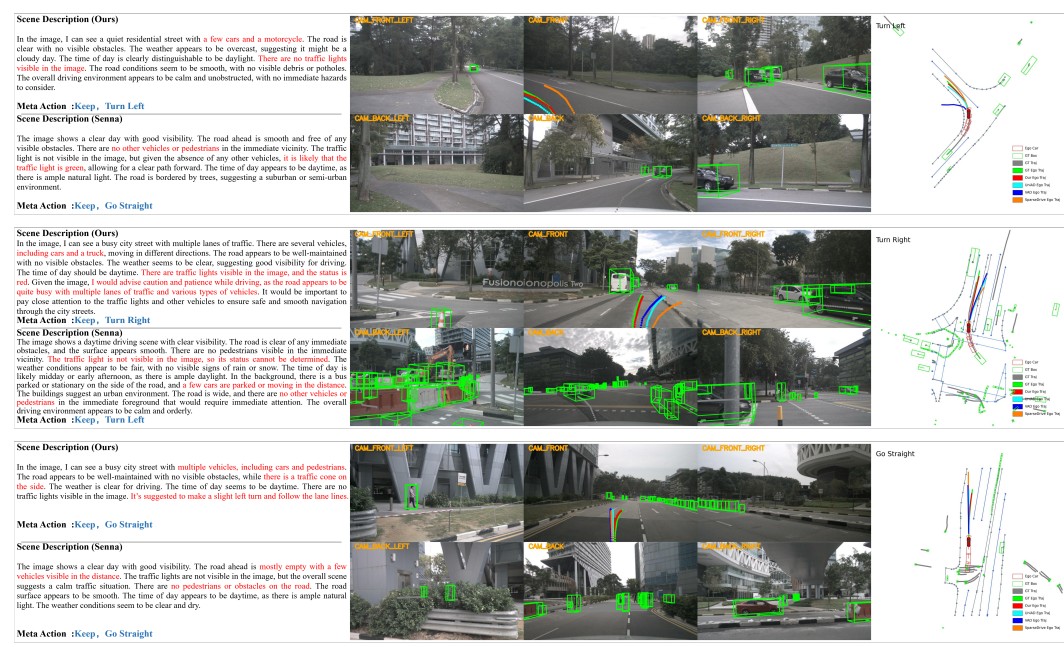

Figure 9: Qualitative results of end-to-end planning and VLM analysis on the nuScenes Caesar et al. (2020) dataset. Our method performs more robust planning under different driving scenarios with enhanced comprehension by scene flow dynamics.

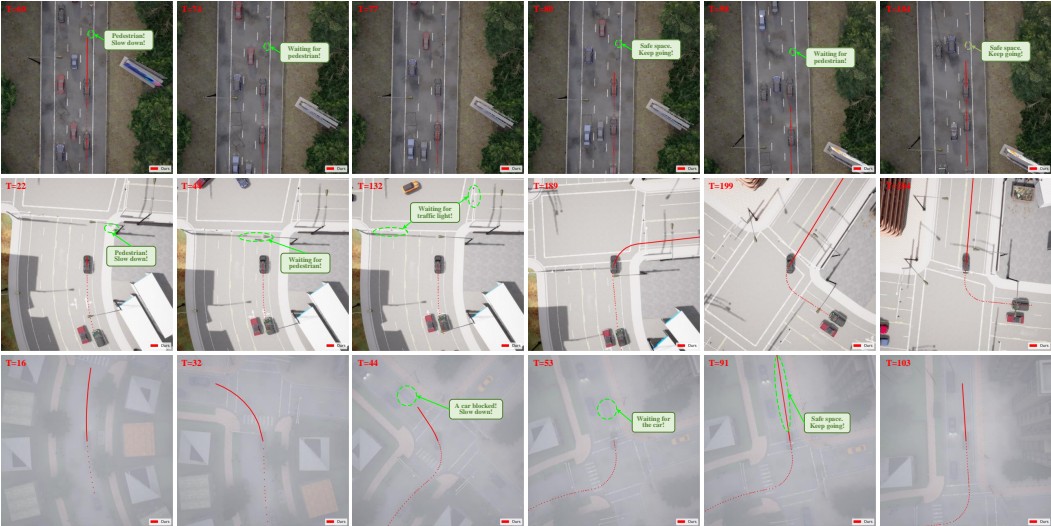

Figure 10: Visualization of the planning results on the Bench2Drive dataset Jia et al. (2024) with CARLA simulator.

Fig. 14. It shows that the standard image features fail to observe the crucial vehicles, which could potentially lead to serious traffic accidents. In contrast, the features with our ego-scene interactive modeling produce a highlighted response to key dynamic targets. Besides, the activation of background clutter is much reduced by our design. These demonstrate that the proposed module successfully guides the model to comprehend the driving environment and focus on critical objects, thereby enhancing the overall robustness and stability of the auto-driving system.

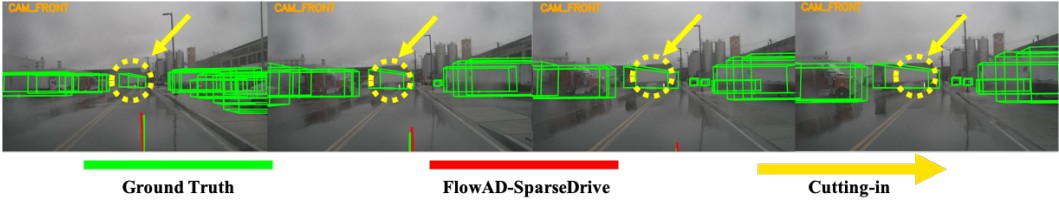

Figure 11: Visualization of a challenging **cutting-in** case. Observing the front vehicle with an abrupt lane change (marked by the yellow arrow), our planner captures the spatio-temporal dynamics and performs the brake, proving the capability to handle complex driving interactions.

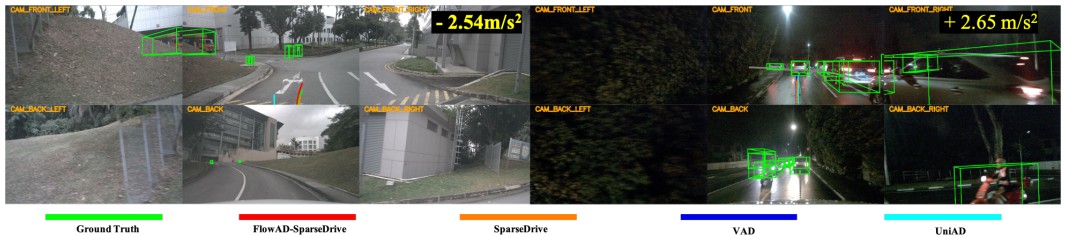

Figure 12: Visualization of planning stability under challenging longitudinal dynamics. Under the complex scenarios including **hard braking** (Left, $-2.54m/s^2$) and **rapid acceleration** (Right, $+2.65m/s^2$), our method is capable of making robust planning compared with other methods.

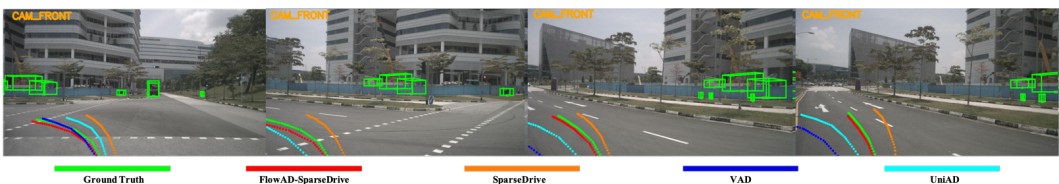

Figure 13: Visualization of planning robustness in a steering case. Our method could plan a more reasonable ego trajectory under the safety-critical turning scenario, proving the effectiveness.

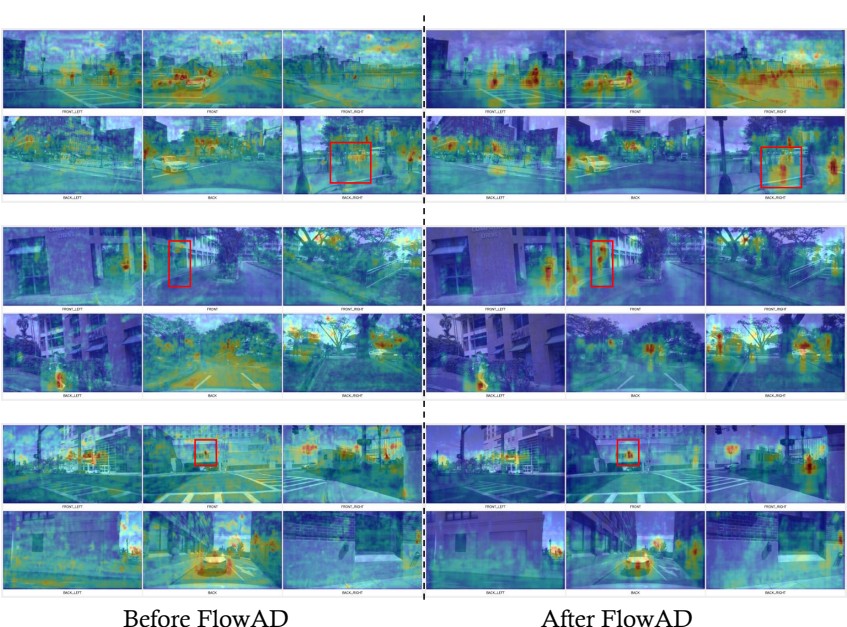

Before FlowAD          After FlowAD

Figure 14: Visualization of activation maps. The proposed flow prediction helps to concentrate on the critical dynamic objects and suppress the background clutter.

