# OpenReview forum: "FlowAD: Ego-Scene Interactive Modeling for Autonomous Driving"
_ICLR.cc/2026/Conference — ICLR 2026 Poster_

### Official Review · Reviewer_17iF · 2025-10-26

**Soundness:** 3
**Presentation:** 4
**Contribution:** 3
**Rating:** 8
**Confidence:** 3

**Summary:**

This paper introduces FlowAD, a novel framework for autonomous driving that explicitly models the ego-vehicle's motion feedback on the perceived environment. The core idea is to represent this interaction as a learnable "scene flow" in the latent feature space. The method is structured around three main components: ego-guided scene partition to create flow units, spatial and temporal flow prediction modules to model dynamics, and task-aware enhancement strategies for downstream tasks. The paper also proposes a new metric, Frames before Correct Planning, to evaluate a planner's scene understanding speed. Extensive experiments on nuScenes and Bench2Drive demonstrate state-of-the-art performance across perception, planning, and VLM-based tasks.

**Strengths:**

1）In this work, the authors introduce a general flow-based framework for autonomous driving. While most existing methods mainly study the planning with current observation, the authors focus on performing the control outputs that shape future sensory input, which is a relatively absence part of existing methods.
2) The proposed FlowAD framework is elegantly designed and demonstrates remarkable generality. According to the experiment, it is successfully applied to diverse baselines (such as SparseBEV, SparseDrive, and Senna) across multiple task types (detection, planning, VLM analysis).
3) The paper is well-written and easy to understand. The experimental section is thorough and convincing.
4) The authors propose a novel and valuable metric, FCP. This new metric addresses a genuine gap in evaluation by quantifying the responsiveness and scene understanding speed of a planner, complementing traditional metrics like L2 error and collision rate.

**Weaknesses:**

1) While the performance are promising,  the proposed method introduces a degradation on the computational cost, as evidenced by the drop in FPS compared to baselines (e.g., SparseDrive 9.0 FPS vs. FlowAD 7.6 FPS with ResNet50). It is recommended to add more detailed discussion on the trade-off between performance and efficiency, and potential avenues for optimization.
2) The concept of "scene flow" is central to the paper. While the intuition is clear, a more precise definition or visualization of what these learned flow features represent in the latent space could strengthen the methodological explanation. For instance, how do these features qualitatively differ from standard BEV or image features?
3) The use of KL divergence loss from world models is presented as a key component for training the flow prediction modules. However, I am curious about the experimental results presented in Table 11. It shows that removing both spatial and temporal losses leads to performance worse than the baseline. This suggests that architecture without proper supervision might be harmful. It is recommended to provide a deeper analysis of why this happens and how critical the specific world model loss formulation is.

**Questions:**

1)The dynamic adjustment of partition size relies on fitting a circle to past ego positions. How robust is this method to noisy or high-frequency steering inputs? Did the authors experiment with simpler or more robust ways to incorporate steering velocity?
2)It seems that the local aggregation uses a fixed range of 3 flow units. Was any other adaptive mechanism considered for this aggregation to handle objects of varying sizes more effectively?
3)The paper mentions using "mechanisms from world models" and the KL loss. Could the authors elaborate on how their spatial and temporal prediction modules differ from a standard recurrent world model applied patch-wise? The novelty in the prediction architecture itself could be clarified.

---

> ### Author Response · Authors · 2025-11-23
> **(Part1/3) Response to Reviewer #17iF**
>
> **Q1:** Performance-efficiency trade-off needs deeper analysis and optimization discussion.
>
> **R1:** Thanks for the helpful and innovative suggestion. As detailed in the ablation studies in Tab. 7 of the manuscript, the multi-level flow prediction is the primary bottleneck causing the FPS degradation. To optimize this trade-off, we perform the ego-scene interactive modeling only on the last stage of multi-view features. The result is listed in Table 1. It shows that the lightweight FlowAD maintains its leading performance while incurring only a minimal drop of 0.5 FPS, achieving a balance between effectiveness and efficiency. Besides reducing the modeled feature levels, increasing the partition size (i.e., decreasing the number of flow units) is another way for the performance-efficiency trade-off, as revealed in Tab. 10 of the appendix. We will explore more strategies in future work, thanks!
>
> We have added the above analysis and experiment in Sec. A.4 of the appendix. Again, thanks!
>
> % This successfully achieves a better balance between accuracy and computational efficiency. In fact, users can dynamically adjust the backbone stage that need to compute scene flow according to their own computing resources.
>
> **Table 1: Results of perception, motion prediction and planning on the nuScenes validation set.**
> *The baseline method is SparseDrive [1]. $^*$ denotes using the last FPN feature for multi-level partition and spatial-temporal flow prediction.*
>
> | Method | Backbone | Det mAP↑ | Det NDS↑ | Trk AMOTA↑ | Trk AMOTP↓ | Map mAP↑ | Mot minADE↓ | Mot minFDE↓ | Plan Avg.L2↓ | Plan Avg.Col↓ | Plan FCP↓ | FPS↑ |
> | :--- | :--- | :--- | :--- | :--- | :--- | :--- | :--- | :--- | :--- | :--- | :--- | :--- |
> | SparseDrive [1] | ResNet50 | 0.418 | 0.525 | 0.386 | 1.254 | 0.551 | 0.62 | 0.99 | 0.61 | 0.08 | 2.55 | **9.0** |
> | **FlowAD (Ours)** | **ResNet50** | **0.448** | **0.555** | **0.427** | **1.181** | **0.582** | **0.58** | **0.95** | **0.56** | **0.06** | **1.03** | 7.6 |
> | **FlowAD $^*$ (Ours)** | **ResNet50** | 0.434 | 0.544 | 0.416 | 1.198 | 0.567 | 0.59 | 0.96 | 0.57 | 0.06 | 1.45 | 8.5 |
>
>
> **Q2:** Qualitative visualization of learned scene flow features.
>
> **R2:** We appreciate this constructive comment. Following the suggestion, we visualize the activation maps of multi-view image features before and after the flow prediction using GradCAM, as shown in Fig. 12 of the appendix. It shows that the standard image features fail to observe the crucial vehicles, which could potentially lead to serious traffic accidents. In contrast, the features with our ego-scene interactive modeling produce a highlighted response to key dynamic targets. Besides, the activation of background clutter is much reduced by our design. These demonstrate that the proposed module successfully guides the model to comprehend the driving environment and focus on critical objects, thereby enhancing the overall robustness and stability of the auto-driving system.
>
> We have added the above analysis and visualization in Sec. A.19 of the appendix, thanks!
>
> **Q3:** Analysis of performance degradation without supervision of world model losses.
>
> **R3:** Thanks for the helpful and innovative suggestion. The KL divergence loss serves as an intrinsic component from the seminal world model (i.e., Dreamer series) to current autonomous driving with world models (e.g., DriveWM [2] and DriveDreamer [3]), which provides crucial supervision for the dreamed future states. Therefore, the performance degradation without the world model loss is foreseeable.
>
> However, the lack of explicit supervision in the world model does not necessarily imply that our ego-scene interactive modeling is detrimental. As shown in Table 2 below, our model without world model loss benefits more from the training epochs compared with the baseline, which achieves superior performance at the training epoch 30. In contrast, the one with explicit supervision reaches the convergence much faster. These reveal that the world model loss helps the model to learn realistic future driving environment, as well as the optimization potential of our design. We will further explore the inherent mechanism in future work, thanks!
>
> We have added the above analysis and experiment in Sec. A.7 of the appendix. Again, thanks!

---

> ### Author Response · Authors · 2025-11-23
> **(Part2/3) Response to Reviewer #17iF**
>
> **Table 2: Ablation on the flow prediction losses on the nuScenes [4] validation set.**
>
> | # | Method | Epoch | Loss Spatial | Loss Temporal | mAP | NDS | mATE | mASE | mAOE | mAVE | mAAE |
> | :---: | :--- | :---: | :---: | :---: | :---: | :---: | :---: | :---: | :---: | :---: | :---: |
> | 1 | SparseBEV [5] | 18 | | | 0.437 | 0.541 | 0.625 | 0.283 | 0.401 | 0.255 | 0.191 |
> | 2 | SparseBEV [5] | 24 | | | 0.445 | 0.553 | 0.605 | 0.278 | 0.389 | 0.253 | 0.189 |
> | 3 | SparseBEV [5] | 30 | | | 0.448 | 0.554 | 0.599 | 0.273 | 0.389 | 0.256 | 0.193 |
> | 4 | FlowAD (Ours) | 18 | | | 0.393 | 0.499 | 0.671 | 0.294 | 0.404 | 0.275 | 0.198 |
> | 5 | FlowAD (Ours) | 24 | | | 0.444 | 0.548 | 0.619 | 0.280 | 0.392 | 0.256 | 0.192 |
> | 6 | FlowAD (Ours) | 30 | | | 0.468 | 0.570 | 0.598 | 0.269 | 0.390 | 0.245 | 0.189 |
> | **7** | **FlowAD (Ours)** | **18** | **✓** | **✓** | **0.454** | **0.557** | **0.611** | **0.270** | **0.385** | **0.244** | **0.187** |
> | **8** | **FlowAD (Ours)** | **24** | **✓** | **✓** | **0.475** | **0.574** | **0.585** | **0.265** | **0.361** | **0.236** | **0.184** |
> | **9** | **FlowAD (Ours)** | **30** | **✓** | **✓** | **0.483** | **0.580** | **0.578** | **0.263** | **0.355** | **0.233** | **0.183** |
>
> **Q4:** Robustness to noisy or high-frequency steering inputs.
>
> **R4:** Thanks for the helpful and innovative suggestion. We evaluate the robustness by (1) injecting Gaussian noise into the observed images, (2) perturbing camera intrinsics/extrinsics and ego-pose parameters by multiplying a random coefficient with the range of 0.95~1.05, and (3) introducing temporal drift by randomly sampling the input frame from $t-2$ to $t$. The results in Table 3 show that our method consistently demonstrates stronger robustness compared with the baseline SparseDrive. In addition, the visualization in Fig. 10/11 of the appendix further substantiates the model's stability under acceleration, deceleration, and steering driving scenarios. We will further explore simpler and more reliable ego-motion integration strategies in future work, thanks!
>
> We have added the above analysis, experiment, and visualization in Sec. A.15/18 of the appendix. Again, thanks!
>
> **Table 3: Robustness analysis on the nuScenes [4] validation set. The baseline method is SparseDrive [1] with ResNet101.**
>
> | Method | Gaussian Noise | Calib. Error | Time Drift | Det mAP | Det NDS | Trk AMOTA | Trk AMOTP | Map mAP | MP minADE | MP minFDE | Plan Avg.L2 | Plan Avg.Col | Plan FCP |
> | :--- | :---: | :---: | :---: | :--- | :--- | :--- | :--- | :--- | :--- | :--- | :--- | :--- | :--- |
> | SparseDrive [1] | | | | 0.496 | 0.588 | 0.501 | 1.085 | 0.562 | 0.60 | 0.96 | 0.58 | 0.06 | 2.30 |
> | **FlowAD (Ours)** | | | | **0.523** | **0.605** | **0.518** | **1.040** | **0.595** | **0.56** | **0.93** | **0.52** | **0.05** | **0.91** |
> | SparseDrive [1] | ✓ | | | 0.472 | 0.555 | 0.487 | 1.159 | 0.533 | 0.63 | 1.03 | 0.60 | 0.07 | 2.34 |
> | **FlowAD (Ours)** | ✓ | | | **0.517** | **0.596** | **0.514** | **1.053** | **0.583** | **0.57** | **0.96** | **0.53** | **0.05** | **0.96** |
> | SparseDrive [1] | | ✓ | | 0.438 | 0.540 | 0.466 | 1.211 | 0.509 | 0.64 | 1.07 | 0.65 | 0.08 | 3.22 |
> | **FlowAD (Ours)** | | ✓ | | **0.486** | **0.570** | **0.498** | **1.103** | **0.543** | **0.58** | **0.99** | **0.56** | **0.06** | **1.55** |
> | SparseDrive [1] | | | ✓ | 0.470 | 0.565 | 0.483 | 1.107 | 0.528 | 0.61 | 1.00 | 0.60 | 0.07 | 3.93 |
> | **FlowAD (Ours)** | | | ✓ | **0.505** | **0.590** | **0.508** | **1.056** | **0.569** | **0.56** | **0.95** | **0.53** | **0.05** | **1.98** |
> | SparseDrive [1] | ✓ | ✓ | ✓| 0.419 | 0.531 | 0.457 | 1.244 | 0.485 | 0.65 | 1.10 | 0.66 | 0.08 | 4.20 |
> | **FlowAD (Ours)** | ✓ | ✓ | ✓ | **0.465** | **0.558** | **0.476** | **1.119** | **0.520** | **0.60** | **1.02** | **0.57** | **0.06** | **2.17** |
>
> **Q5:** Try other adaptive aggregation mechanisms to handle objects of varying sizes.
>
> **R5:** Thanks for the helpful and innovative suggestion. As suggested, we exploit the Deformable Attention to dynamically aggregate the surrounding environmental information for each partitioned patch. Notably, the initial sampling locations are uniformly distributed in local 3 flow units, which are then dynamically adjusted by the predicted offsets. Considering the computation costs caused by the high resolution of features in earlier FPN stages, the aggregation is only performed on the last two stages. The result is listed in Table 4 below. It shows that the mAP/NDS further improves, while the FPS degrades for 2.2 FPS. We will explore more efficient dynamic aggregation strategy in future work, thanks!
>
> We have added the above analysis and experiment in Sec. A.9 of the appendix. Again, thanks!

---

> ### Author Response · Authors · 2025-11-23
> **(Part3/3) Response to Reviewer #17iF**
>
> **Table 4: Ablation on the range of local aggregation on the nuScenes [4] validation set.**
>
> | Method | Aggregation | mAP | NDS | mATE | mASE | mAOE | mAVE | mAAE | FPS |
> | :--- | :--- | :--- | :--- | :--- | :--- | :--- | :--- | :--- | :--- |
> | SparseBEV [5] | | 0.445 | 0.553 | 0.605 | 0.278 | 0.389 | 0.253 | 0.189 | 21.7 |
> | **FlowAD (Ours)** | **Local** | 0.475 | 0.574 | 0.585 | **0.265** | **0.361** | **0.236** | **0.184** | **18.9** |
> | FlowAD (Ours) | Deformable | **0.480** | **0.580** | **0.582** | 0.273 | 0.367 | 0.241 | 0.185 | 16.7 |
>
> **Q6:** Difference between spatial and temporal flow prediction modules and a standard recurrent world model applied patch-wise.
>
> **R6:** We appreciate this helpful and constructive suggestion. The difference lies in three aspects: (1) Extra spatial prediction. Besides the well-known temporal prediction of future driving scenarios, we also perform the forecasting on the spatial dimension. The spatial flow prediction module captures the dynamics from ahead flow units and forecasts the rear ones, which helps the model better understand the scene flow caused by ego motion (proved in the ablation of Tab. 7 in the manuscript). (2) Introduction of ego-motion influence. The ego motion directly decides the modeled areas of patches with the starting point and size of the partition. This endows the model with better capability of comprehending the ego-scene interactive process. (3) Decoupling of transition state and environmental feature. Different from standard world model that regards the constantly-updated transition state as the predicted environmental feature, we decouple the two concepts to release the modeling potential respectively. Specifically, the transition state modeled with GRU (same as the standard world model) is exploited to backward update the observation feature with cross-attention (Fig. 3 of the manuscript). The ablation in Tab. 10 of the appendix further proves our claim that the decoupling helps to better extract environmental information and benefit planning.
>
> We have added the above analysis in Sec. A.16 of the appendix, thanks!
>
>
> ## References
> [1] Sparsedrive: End-to-end autonomous driving via sparse scene representation. arXiv:2405.19620, 2024.
>
> [2] Driving into the future: Multiview visual forecasting and planning with world model for autonomous driving. In CVPR, 2024.
>
> [3] Drivedreamer: Towards real-world-drive world models for autonomous driving. In ECCV, 2024.
>
> [4] Nuscenes: A multimodal dataset for autonomous driving. In CVPR, 2020.
>
> [5] Sparsebev: High-performance sparse 3d object detection from multi-camera videos. In ICCV, 2023.

---

### Official Review · Reviewer_n4vr · 2025-10-27

**Soundness:** 4
**Presentation:** 3
**Contribution:** 3
**Rating:** 8
**Confidence:** 4

**Summary:**

This paper introduces FlowAD, a unified ego-scene interactive modeling framework for autonomous driving. Unlike standard BEV-based approaches that mostly treat ego-motion as an external factor to compensate for, FlowAD attempts to integrate ego trajectory information directly into the representation space through a flow-unit partitioning scheme. The method includes an ego-guided dynamic partition, spatial-temporal flow prediction, and a task-aware enhancement module, which together aim to improve perception and planning by better capturing the interaction between the ego vehicle and the surrounding scene. A new evaluation metric, FCP (Flow Consistency Precision), is proposed to measure alignment between ego-motion and scene evolution. Experiments on standard benchmarks show consistent improvements over SparseDrive.

**Strengths:**

I appreciate the clarity with which the authors formulate ego-scene interaction as a modeling problem. This is a conceptual contribution with potential impact. The flow-unit partitioning is elegant and physically interpretable. The method’s modular design allows integration into various downstream tasks, which enhances its practical value. The introduction of FCP as a metric is also interesting, as it focuses on ego-scene alignment, something that existing metrics often overlook. Empirical results are consistent across multiple tasks, lending credibility to the claims.

**Weaknesses:**

Robustness is also under-characterized: the method hinges on ego-guided partition but offers little evidence for stability under sharp turns, high speeds, or other extreme maneuvers. The proposed FCP metric is interesting yet its external validity remains unclear—there is no systematic analysis of how it correlates with closed-loop planning metrics across routes/seeds or datasets.

**Questions:**

How does FlowAD compare with a token-based BEV fusion baseline and a compact world-model/diffusion-style planner under matched training and evaluation conditions?

How does FCP correlate with standard closed-loop planning metrics such as driving score, collision rate, and comfort across multiple runs or datasets?

---

> ### Author Response · Authors · 2025-11-23
> **(Part1/3) Response to Reviewer #n4vr**
>
> **Q1:** Insufficient robustness analysis under extreme maneuvers.
>
> **R1:** Thanks for the helpful and innovative suggestion.
> We address this concern by evaluating the metrics of different driving scenes in Table 1 below. According to the given driving command (*i.e.*, *go straight*, *turn left* and *turn right*), we divide the 6,019 validation samples in nuScenes [1] into three parts, which contain 5,309, 301 and 409 ones, respectively. Not surprisingly, all methods perform better under *go straight* scenes than the steering scenes. When applying the proposed ego-scene interactive modeling, our planners achieve considerable gains in steering scenes, proving its robustness. Furthermore, the ability metrics on more challenging Bench2Drive [2] (Tab. 5 of the manuscript) also confirm the stability of our approach in diverse and extreme maneuver scenarios. The case presented in Fig. 9/10/11 of the appendix also verifies this with safer planning compared with other methods.
>
> We have updated the above analysis, experiment, and visualization in Sec. A.13/A.18 of the appendix, thanks!
>
> **Table 1: Performance comparison under different driving scenarios on the nuScenes validation set.**
> *Sample counts: Go Straight (5309), Turn Left (301), Turn Right (409), Overall (6019).*
>
> | Method | Backbone | Straight L2 (m)↓ | Straight Col. (%)↓ | Left L2 (m)↓ | Left Col. (%)↓ | Right L2 (m)↓ | Right Col. (%)↓ | Overall L2 (m)↓ | Overall Col. (%)↓ |
> | :--- | :--- | :--- | :--- | :--- | :--- | :--- | :--- | :--- | :--- |
> | SparseDrive [3] | ResNet50 | 0.58 | 0.08 | 0.83 | 0.29 | 0.86 | **0.08** | 0.61 | 0.09 |
> | **FlowAD $_{SparseDrive}$** | **ResNet50** | **0.53** | **0.06** | **0.77** | **0.08** | **0.69** | 0.09 | **0.56** | **0.06** |
> | SparseDrive [3] | ResNet101 | 0.56 | 0.06 | 0.89 | 0.13 | 0.81 | 0.10 | 0.58 | 0.06 |
> | **FlowAD $_{SparseDrive}$** | **ResNet101** | **0.50** | **0.05** | **0.74** | **0.08** | **0.65** | **0.08** | **0.52** | **0.05** |
> | DiffusionDrive [4] | ResNet50 | 0.55 | 0.07 | 0.64 | 0.12 | 0.59 | 0.10 | 0.57 | 0.08 |
> | **FlowAD $_{DiffusionDrive}$** | **ResNet50** | **0.54** | **0.03** | **0.62** | **0.02** | **0.54** | **0.07** | **0.54** | **0.06** |
>
>
> **Q2:** How does the FCP metric correlate with closed-loop metrics.
>
> **R2:**
> We appreciate the helpful and innovative suggestion. The reliance on Ground-Truth trajectory is the main impediment to correlating with closed-loop metrics and serving as part of them. To address this limitation and better evaluate reaction speed, we have extended the FCP metric to move beyond simple GT matching. Specifically, we define a ''successful reaction'' based on the trajectory's compliance with the high-level command rather than its proximity to the GT. For a given command (provided in both open- and closed-loop settings), the model is considered responsive if its predicted trajectory at the $3\text{s}$ horizon aligns with the command for a predefined sequence of $q$ frames. For instance, regarding a ``Turn Right'' command, adopting the threshold from UniAD [5], a planner is deemed successful if the lateral displacement of the ego trajectory at the final timestamp exceeds $2.0\text{m}$ for $q$ consecutive frames (please refer to Sec. A.12 of the appendix for the detailed formulation, thanks!).
>
> **Table 2: Planning results in terms of L2 error, collision rate, and the proposed FCP metrics with different thresholds on the nuScenes [1] validation set. The baseline method is SparseDrive [3]. $^{\diamondsuit}$ denotes the refined metric.**
>
> | Method | Backbone | L2 1s | L2 2s | L2 3s | L2 Avg | Col 1s | Col 2s | Col 3s | Col Avg | FCP 0.25m | FCP 0.50m | FCP 0.75m | FCP Avg | FCP $^{\diamondsuit}$ 1frame | FCP $^{\diamondsuit}$ 2frame | FCP $^{\diamondsuit}$ 3frame | FCP $^{\diamondsuit}$ Avg | FPS |
> | :--- | :--- | :--- | :--- | :--- | :--- | :--- | :--- | :--- | :--- | :--- | :--- | :--- | :--- | :--- | :--- | :--- | :--- | :--- |
> | UniAD [5] | ResNet101 | 0.44 | 0.67 | 0.96 | 0.69 | 0.04 | 0.08 | 0.23 | 0.12 | 5.41 | 2.69 | 0.93 | 3.01 | 0.73 | 2.02 | 3.43 | 2.06 | 1.8 |
> | VAD [6] | ResNet50 | 0.41 | 0.70 | 1.05 | 0.72 | 0.07 | 0.17 | 0.41 | 0.22 | 6.44 | 3.07 | 1.42 | 3.64 | 0.87 | 2.39 | 4.12 | 2.46 | 5.2 |
> | MomAD [7] | ResNet50 | 0.31 | 0.57 | 0.91 | 0.60 | 0.01 | 0.05 | 0.22 | 0.09 | - | - | - | - | - | - | - | - | 7.8 |
> | SparseDrive [3] | ResNet50 | 0.29 | 0.58 | 0.96 | 0.61 | 0.01 | 0.05 | 0.18 | 0.08 | 4.87 | 2.55 | 1.09 | 2.84 | 0.54 | 1.88 | 3.02 | 1.48 | **9.0** |
> | **FlowAD (Ours)** | **ResNet50** | **0.28** | **0.55** | **0.84** | **0.56** | **0.01** | **0.03** | **0.15** | **0.06** | **2.12** | **1.03** | **0.48** | **1.21** | **0.37** | **0.90** | **1.79** | **1.02** | 7.6 |

---

> ### Author Response · Authors · 2025-11-23
> **(Part2/3) Response to Reviewer #n4vr**
>
> We then perform comparison using the extended FCP $^{\diamondsuit}$ metric, as shown in Table 2 above. The results indicate that our method consistently maintains its superiority over competing approaches across varying thresholds of $q$, validating the robustness of our planner. Moreover, by eliminating GT dependence and accommodating diverse decision-making behaviors, this refined metric offers the flexibility to be deployed in both open-loop and closed-loop settings. We hope FCP $^{\diamondsuit}$ serves as a standard, robust benchmark for a more holistic assessment of autonomous driving capabilities.
>
> We have added the above analysis and experiment in Sec. A.12 of the appendix. Again, thanks!
>
> **Q3:** Comparison between FlowAD and world-model/diffusion planners.
>
> **R3:** Thanks for the helpful and innovative comment. Following the suggestion, we extend the comparison with a world-model based planner (i.e., DriveWM [8]) and a diffusion-style planner (i.e., DiffusionDrive [4]). Notably, we apply our proposed FlowAD to DiffusionDrive to ensure a fair comparison under matched training and evaluation conditions, i.e., FlowAD-DiffusionDrive. As shown in Table 3 below, our planners demonstrate superior performance compared with other world-model/diffusion-style planners, proving the effectiveness of our ego-scene interactive modeling. We have updated Tab. 4 of the manuscript by adding the above-mentioned methods, as well as recent SOTAs, thanks!
>
> **Table 3: Results of perception, motion prediction and planning on the nuScenes validation set.** The FPS is measured on an RTX3090 GPU. The baselines are SparseDrive and DiffusionDrive. $^{\ddagger}$ denotes using ego status in the planning module.
>
> | Method | Backbone | Det mAP↑ | Det NDS↑ | Trk AMOTA↑ | Trk AMOTP↓ | Map mAP↑ | Mot minADE↓ | Mot minFDE↓ | Plan Avg.L2↓ | Plan Avg.Col↓ | Plan FCP↓ | FPS↑ |
> | :--- | :--- | :--- | :--- | :--- | :--- | :--- | :--- | :--- | :--- | :--- | :--- | :--- |
> | VAD [6] | ResNet50 | - | - | - | - | 0.476 | - | - | 0.72 | 0.21 | 3.07 | 5.2 |
> | MomAD [7] | ResNet50 | 0.423 | 0.531 | 0.391 | 1.243 | 0.559 | 0.61 | 0.98 | 0.60 | 0.09 | - | 7.8 |
> | DriveWM [8] | - | - | - | - | - | - | - | - | 0.80 | 0.26 | - | - |
> | LAW [9] | - | - | - | - | - | - | - | - | 0.61 | 0.30 | - | - |
> | UncAD [10] | ResNet50 | - | - | - | - | - | - | - | 0.60 | 0.07 | - | - |
> | FocalAD [11] | ResNet50 | - | - | - | - | - | 0.61 | 0.95 | 0.60 | 0.09 | - | - |
> | GenAD [12] | ResNet50 | 0.290 | - | - | - | - | - | - | 0.58 | 0.16 | 1.78 | 6.7 |
> | ORION (VLM) [13] | - | - | - | - | - | - | - | - | 0.63 | 0.37 | - | - |
> | SparseDrive [3] | ResNet50 | 0.418 | 0.525 | 0.386 | 1.254 | 0.551 | 0.62 | 0.99 | 0.61 | 0.08 | 2.55 | **9.0** |
> | **FlowAD $_{SparseDrive}$ (Ours)** | **ResNet50** | 0.448 | 0.555 | 0.427 | 1.181 | 0.582 | **0.58** | **0.95** | 0.56 | 0.06 | **1.03** | 7.6 |
> | DiffusionDrive [4] | ResNet50 | 0.412 | 0.522 | 0.374 | 1.246 | 0.559 | 0.64 | 1.01 | 0.57 | 0.08 | 1.35 | 6.8 |
> | **FlowAD $_{DiffusionDrive}$ (Ours)** | **ResNet50** | **0.477** | **0.567** | **0.466** | **1.141** | **0.605** | 0.61 | 0.97 | **0.54** | **0.04** | 1.05 | 5.4 |
> | UniAD [5] | ResNet101 | 0.380 | 0.498 | 0.359 | 1.320 | - | 0.71 | 1.02 | 0.69 | 0.12 | 2.69 | 1.8 |
> | SparseDrive [3] | ResNet101 | 0.496 | 0.588 | 0.501 | 1.085 | 0.562 | 0.60 | 0.96 | 0.58 | 0.06 | 2.30 | **7.3** |
> | **FlowAD $_{SparseDrive}$ (Ours)** | **ResNet101** | **0.523** | **0.605** | **0.518** | **1.040** | **0.595** | **0.56** | **0.93** | **0.52** | **0.05** | **0.91** | 5.4 |
> | DriveDreamer $^{\ddagger}$ [14] | - | - | - | - | - | - | - | - | 0.29 | 0.15 | - | - |
> | SSR $^{\ddagger}$ [15] | ResNet50 | - | - | - | - | - | - | - | 0.39 | 0.06 | 1.91 | **19.6** |
> | DriveTransformer $^{\ddagger}$ [16]  | ResNet50 | - | - | - | - | - | - | - | 0.40 | 0.11 | - | - |
> | **FlowAD $_{SparseDrive}^{\ddagger}$ (Ours)** | **ResNet50** | **0.451** | **0.554** | **0.430** | **1.198** | **0.576** | **0.57** | **0.94** | **0.25** | **0.04** | **0.75** | 7.6 |

---

> ### Author Response · Authors · 2025-11-23
> **(Part3/3) Response to Reviewer #n4vr**
>
> **Q4:** Combine FCP with closed-loop metrics across multiple runs/datasets.
>
> **R4:** Thanks for the constructive and helpful suggestion. As discussed in Q2-R2, the extended FCP metric is Ground Truth (GT)-independent and evaluates the model's reaction speed based on generally available high-level commands. Therefore, it's natural to insert FCP as a constituent component of the overall Driving Score, e.g., performing a weighted sum with other sub-metrics in the closed-loop evaluation. Besides, the collision and emergency brake/rapid acceleration could be exploited as the penalty coefficient in the FCP metric. We will explore more reasonable computation pipelines to assess the reaction speed of auto-driving systems in future work. Again, thanks!
>
>
> ## References
> [1] Nuscenes: A multimodal dataset for autonomous driving. In CVPR, 2020.
>
> [2] Bench2drive: Towards multi-ability benchmarking of closed-loop end-to-end autonomous driving. arXiv:2406.03877, 2024.
>
> [3] Sparsedrive: End-to-end autonomous driving via sparse scene representation. arXiv:2405.19620, 2024.
>
> [4] Diffusiondrive: Truncated diffusion model for end-to-end autonomous driving. In CVPR, 2025.
>
> [5] Planning-oriented autonomous driving. In CVPR, 2023.
>
> [6] Vad: Vectorized scene representation for efficient autonomous driving. arXiv:2303.12077, 2023.
>
> [7] Don’t shake the wheel: Momentum-aware planning in end-to-end autonomous driving. arXiv:2503.03125, 2025.
>
> [8] Driving into the future: Multiview visual forecasting and planning with world model for autonomous driving. In CVPR, 2024.
>
> [9] Enhancing end-to-end autonomous driving with latent world model. arXiv:2406.08481, 2024.
>
> [10] Uncad: Towards safe end-to-end autonomous driving via online map uncertainty. arXiv:2504.12826, 2025.
>
> [11] Focalad: Local motion planning for end-to-end autonomous driving. arXiv:2506.11419, 2025.
>
> [12] Genad: Generative end-to-end autonomous driving. In ECCV, 2024.
>
> [13] Orion: A holistic end-to-end autonomous driving framework by vision-language instructed action generation. arXiv:2503.19755, 2025.
>
> [14] Drivedreamer: Towards real-world-drive world models for autonomous driving. In ECCV, 2024.
>
> [15] Navigation-guided sparse scene representation for end-to-end autonomous driving. arXiv:2409.18341, 2024.
>
> [16] Drivetransformer: Unified transformer for scalable end-to-end autonomous driving. arXiv:2503.07656, 2025.

---

> > ### Comment · Reviewer_n4vr · 2025-11-26
> >
> > Excellent work. The author's rebuttal resolved most of my concerns, and I will maintain my rating.

---

### Official Review · Reviewer_8pZR · 2025-10-27

**Soundness:** 2
**Presentation:** 2
**Contribution:** 2
**Rating:** 6
**Confidence:** 3

**Summary:**

This paper proposes FlowAD, a general flow-based framework for autonomous driving that addresses a key limitation in existing approaches: the inadequate modeling of ego motion feedback into scene observation. Drawing inspiration from human perception and relative motion, FlowAD introduces an ego-scene interactive modeling paradigm, representing ego-scene interaction as scene flow relative to the ego vehicle. The methodology centers around an ego-guided scene partition, spatial and temporal flow prediction modules, and task-aware enhancement for downstream perception, planning, and VLM  analysis tasks. A new evaluation metric, FCP, is proposed to quantify scene understanding. The framework is evaluated across several benchmarks and tasks, demonstrating consistent gains over baselines on multiple metrics.

**Strengths:**

- The paper explicitly addresses the gap of neglecting ego vehicle feedback by proposing to model this effect as a “relative scene flow,” balancing physical intuition with practical data-driven learning from logged data.
- FlowAD is built with end-to-end components, making it pluggable into various autonomous driving baselines.
- Extensive experiments covering open-loop and closed-loop scenarios, multiple tasks, ablation studies, and introducing the FCP metric, effectively demonstrate the method's efficacy and its potential advantages over the baselines.

**Weaknesses:**

- Many world-model–style approaches like DriveDreamer [1] and Drive-WM [2] exploit action-guided future status to optimize the trajectory. The claim that previous methods “ignore feedback” seems somewhat absolute and would benefit from a more rigorous comparison and clear delineation of the differences.
- The estimation of the partition starting point and turning radius depends heavily on the accuracy of the ego vehicle’s pose/odometry. It would be better to give more analysis of sensitivity to noise, calibration errors, or timestamp drift. Besides, partitioning exclusively along the image width assumes that the dominant relative motion is horizontal. The method’s performance under conditions with significant changes in longitudinal dynamics (e.g., on slopes or during acceleration/deceleration) or notable camera pitch variations is not analyzed.
- The chosen baseline methods are primarily from 2023 to 2024, and it would be beneficial to include comparisons with more recent approaches, such as the perception-free methods SSR [3] and LAW [4].

[1] DriveDreamer: Towards Real-world-driven World Models for Autonomous Driving. arXiv. 2309.09777.

[2] Driving into the Future: Multiview Visual Forecasting and Planning with World Model for Autonomous Driving. arXiv. 2311.17918.

[3] Navigation-guided sparse scene representation for end-to-end autonomous driving. arXiv:2409.18341.

[4] Enhancing end-to-end autonomous driving with latent world model. arXiv preprint arXiv:2406.08481.

**Questions:**

- Could the authors provide a more detailed comparison and discussion with world-model–based methods to clarify the statement made in the introduction? Furthermore, could comparisons with more recent approaches, such as the perception-free methods SSR and LAW, be included to strengthen the evaluation?
- How sensitive is the final performance to the chosen partition size and the parameters within the dynamic adjustment module? In particular, what is the model's robustness in the presence of inaccurate ego-motion estimates?

---

> ### Author Response · Authors · 2025-11-23
> **(Part1/3) Response to Reviewer #8pZR**
>
> **Q1:** Comparison between FlowAD and world-model-style approaches.
>
> **R1:** Thanks for the helpful and innovative suggestion. We have revised the somewhat absolute expression 'ignore feedback' in the caption of Fig. 1 in the manuscript. DriveDreamer [1] and Drive-WM [2] regard the whole driving environment as a single feature embedding and directly predict the future state driven by the action embedding. Although achieving remarkable driving performance, it neglects the fine-grained environmental messages and the physical influence of ego motion on the camera observations. In contrast, our method divides the driving scenario into multiple parts and extracts the dynamics by flow units. The feedback of ego-motion is explicitly captured by adjusting the start point and ego-left/right sizes of the flow unit partition. These make our ego-scene interactive modeling more sensitive to surrounding dynamics and help the planner understand the driving process and predict robust ego trajectories. In addition, our scene flow performs on both temporal and spatial dimensions, which further benefits the spatial perception of the environment (Tab. 7 of the manuscript). The performance comparison with the mentioned world-model-style methods in Tab. 4 of the manuscript also corroborates these advantages.
>
> We have added the above analysis in Sec. A.16 of the appendix, thanks!
>
> **Q2:** FlowAD's sensitivity to noise, calibration errors, or timestamp drift and robustness under large longitudinal dynamics changes.
>
> **R2:** We appreciate this constructive suggestion.
> Following the reviewer's advice, we evaluate robustness by (1) injecting Gaussian noise into the observed images, (2) perturbing camera intrinsics/extrinsics and ego-pose parameters by multiplying a random coefficient with the range of 0.95~1.05, and (3) introducing temporal drift by randomly sampling the input frame from $t-2$ to $t$. The results in Table 1 show that our method consistently demonstrates stronger robustness compared with the baseline SparseDrive. In addition, the visualization in Fig. 10 of the appendix further substantiates the model's stability under acceleration, deceleration, and uphill/downhill driving scenarios.
>
> We have added the above analysis, experiment, and visualization in Sec. A.15/A.18 of the appendix. Again, thanks!
>
> **Table 1: Robustness analysis on the nuScenes [3] validation set. The baseline method is SparseDrive [4] with ResNet101.**
>
> | Method | Gaussian Noise | Calib. Error | Time Drift | Det mAP | Det NDS | Trk AMOTA | Trk AMOTP | Map mAP | MP minADE | MP minFDE | Plan Avg.L2 | Plan Avg.Col | Plan FCP |
> | :--- | :---: | :---: | :---: | :--- | :--- | :--- | :--- | :--- | :--- | :--- | :--- | :--- | :--- |
> | SparseDrive [4] | | | | 0.496 | 0.588 | 0.501 | 1.085 | 0.562 | 0.60 | 0.96 | 0.58 | 0.06 | 2.30 |
> | **FlowAD (Ours)** | | | | **0.523** | **0.605** | **0.518** | **1.040** | **0.595** | **0.56** | **0.93** | **0.52** | **0.05** | **0.91** |
> | SparseDrive [4] | ✓ | | | 0.472 | 0.555 | 0.487 | 1.159 | 0.533 | 0.63 | 1.03 | 0.60 | 0.07 | 2.34 |
> | **FlowAD (Ours)** | ✓ | | | **0.517** | **0.596** | **0.514** | **1.053** | **0.583** | **0.57** | **0.96** | **0.53** | **0.05** | **0.96** |
> | SparseDrive [4] | | ✓ | | 0.438 | 0.540 | 0.466 | 1.211 | 0.509 | 0.64 | 1.07 | 0.65 | 0.08 | 3.22 |
> | **FlowAD (Ours)** | | ✓ | | **0.486** | **0.570** | **0.498** | **1.103** | **0.543** | **0.58** | **0.99** | **0.56** | **0.06** | **1.55** |
> | SparseDrive [4] | | | ✓ | 0.470 | 0.565 | 0.483 | 1.107 | 0.528 | 0.61 | 1.00 | 0.60 | 0.07 | 3.93 |
> | **FlowAD (Ours)** | | | ✓ | **0.505** | **0.590** | **0.508** | **1.056** | **0.569** | **0.56** | **0.95** | **0.53** | **0.05** | **1.98** |
> | SparseDrive [4] | ✓ | ✓ | ✓ | 0.419 | 0.531 | 0.457 | 1.244 | 0.485 | 0.65 | 1.10 | 0.66 | 0.08 | 4.20 |
> | **FlowAD (Ours)** | ✓ | ✓ | ✓ | **0.465** | **0.558** | **0.476** | **1.119** | **0.520** | **0.60** | **1.02** | **0.57** | **0.06** | **2.17** |

---

> ### Author Response · Authors · 2025-11-23
> **(Part2/3) Response to Reviewer #8pZR**
>
> **Q3:** Lack of comparison with recent methods.
>
> **R3:** Thanks for the helpful and innovative suggestion.
> Following the reviewers’ suggestions, we have incorporated more recent methods for comparison and further applied our approach to DiffusionDrive [5]. As shown in Table 2, our planning scheme still demonstrates outstanding performance compared to the perception-free methods SSR [6] and LAW [7], which validates its effectiveness for a deeper understanding of the driving process. We have updated Tab. 4 of the manuscript by adding the above-mentioned methods, as well as recent SOTAs, thanks!
>
> **Table 2: Results of perception, motion prediction and planning on the nuScenes validation set.** The FPS is measured on an RTX3090 GPU. The baselines are SparseDrive and DiffusionDrive. $^{\ddagger}$ denotes using ego status in the planning module.
>
> | Method | Backbone | Det mAP↑ | Det NDS↑ | Trk AMOTA↑ | Trk AMOTP↓ | Map mAP↑ | Mot minADE↓ | Mot minFDE↓ | Plan Avg.L2↓ | Plan Avg.Col↓ | Plan FCP↓ | FPS↑ |
> | :--- | :--- | :--- | :--- | :--- | :--- | :--- | :--- | :--- | :--- | :--- | :--- | :--- |
> | VAD [8] | ResNet50 | - | - | - | - | 0.476 | - | - | 0.72 | 0.21 | 3.07 | 5.2 |
> | MomAD [9] | ResNet50 | 0.423 | 0.531 | 0.391 | 1.243 | 0.559 | 0.61 | 0.98 | 0.60 | 0.09 | - | 7.8 |
> | DriveWM [2] | - | - | - | - | - | - | - | - | 0.80 | 0.26 | - | - |
> | LAW [7] | - | - | - | - | - | - | - | - | 0.61 | 0.30 | - | - |
> | UncAD [10] | ResNet50 | - | - | - | - | - | - | - | 0.60 | 0.07 | - | - |
> | FocalAD [11] | ResNet50 | - | - | - | - | - | 0.61 | 0.95 | 0.60 | 0.09 | - | - |
> | GenAD [12] | ResNet50 | 0.290 | - | - | - | - | - | - | 0.58 | 0.16 | 1.78 | 6.7 |
> | ORION (VLM) [13] | - | - | - | - | - | - | - | - | 0.63 | 0.37 | - | - |
> | SparseDrive [4] | ResNet50 | 0.418 | 0.525 | 0.386 | 1.254 | 0.551 | 0.62 | 0.99 | 0.61 | 0.08 | 2.55 | **9.0** |
> | **FlowAD $_{SparseDrive}$ (Ours)** | **ResNet50** | 0.448 | 0.555 | 0.427 | 1.181 | 0.582 | **0.58** | **0.95** | 0.56 | 0.06 | **1.03** | 7.6 |
> | DiffusionDrive [5] | ResNet50 | 0.412 | 0.522 | 0.374 | 1.246 | 0.559 | 0.64 | 1.01 | 0.57 | 0.08 | 1.35 | 6.8 |
> | **FlowAD $_{DiffusionDrive}$ (Ours)** | **ResNet50** | **0.477** | **0.567** | **0.466** | **1.141** | **0.605** | 0.61 | 0.97 | **0.54** | **0.04** | 1.05 | 5.4 |
> | UniAD [14] | ResNet101 | 0.380 | 0.498 | 0.359 | 1.320 | - | 0.71 | 1.02 | 0.69 | 0.12 | 2.69 | 1.8 |
> | SparseDrive [4] | ResNet101 | 0.496 | 0.588 | 0.501 | 1.085 | 0.562 | 0.60 | 0.96 | 0.58 | 0.06 | 2.30 | **7.3** |
> | **FlowAD $_{SparseDrive}$ (Ours)** | **ResNet101** | **0.523** | **0.605** | **0.518** | **1.040** | **0.595** | **0.56** | **0.93** | **0.52** | **0.05** | **0.91** | 5.4 |
> | DriveDreamer $^{\ddagger}$ [1] | - | - | - | - | - | - | - | - | 0.29 | 0.15 | - | - |
> | SSR $^{\ddagger}$ [6] | ResNet50 | - | - | - | - | - | - | - | 0.39 | 0.06 | 1.91 | **19.6** |
> | DriveTransformer $^{\ddagger}$ [15]  | ResNet50 | - | - | - | - | - | - | - | 0.40 | 0.11 | - | - |
> | **FlowAD $_{SparseDrive}^{\ddagger}$ (Ours)** | **ResNet50** | **0.451** | **0.554** | **0.430** | **1.198** | **0.576** | **0.57** | **0.94** | **0.25** | **0.04** | **0.75** | 7.6 |

---

> ### Author Response · Authors · 2025-11-23
> **(Part3/3) Response to Reviewer #8pZR**
>
> **Q4:** Sensitivity to partition size, dynamic adjustment parameters, and robustness to inaccurate ego-motion.
>
> **R4:** Thanks for the helpful and innovative suggestion.
>
> (1) We have ablated different partition sizes in Tab. 9 of the appendix. It shows that patch sizes of ${8,4,2,1}$ for the four-level image features achieve the best performance. An interesting observation is that increasing the partition size would instead degrade the performance. The reason lies in the oversized flow unit, which makes it hard to quantify the scele flow dynamics.
>
> (2) In Sec. 3.1, we choose the quadratic power for the coefficients of ego-left/right partition sizes. Here we further explore the influence by ablating the power, as in Tab. 14 of the appendix. The results show that linear power is not sufficient to reflect the different motion speeds of ego-left/right, which is inferior to the one with quadratic power. Notably, the cubic power is overlarge to balance the partition sizes of flow units on both sides. The orientation-relevant metrics are further improved (e.g., mASE and mAOE), while the overall mAP/NDS is inferior to the quadratic one.
>
> (3) Please refer to the robustness experiment in Q2-R2 for more details.
>
> We have updated the above analysis and experiment in Sec. A.8 of the appendix. Again, thanks!
>
> ## References
> [1] Drivedreamer: Towards real-world-drive world models for autonomous driving. In ECCV, 2024.
>
> [2] Driving into the future: Multiview visual forecasting and planning with world model for autonomous driving. In CVPR, 2024.
>
> [3] Nuscenes: A multimodal dataset for autonomous driving. In CVPR, 2020.
>
> [4] Sparsedrive: End-to-end autonomous driving via sparse scene representation. arXiv:2405.19620, 2024.
>
> [5] Diffusiondrive: Truncated diffusion model for end-to-end autonomous driving. In CVPR, 2025.
>
> [6] Navigation-guided sparse scene representation for end-to-end autonomous driving. arXiv:2409.18341, 2024.
>
> [7] Enhancing end-to-end autonomous driving with latent world model. arXiv:2406.08481, 2024.
>
> [8] Vad: Vectorized scene representation for efficient autonomous driving. arXiv:2303.12077, 2023.
>
> [9] Don’t shake the wheel: Momentum-aware planning in end-to-end autonomous driving. arXiv:2503.03125, 2025.
>
> [10] Uncad: Towards safe end-to-end autonomous driving via online map uncertainty. arXiv:2504.12826, 2025.
>
> [11] Focalad: Local motion planning for end-to-end autonomous driving. arXiv:2506.11419, 2025.
>
> [12] Genad: Generative end-to-end autonomous driving. In ECCV, 2024.
>
> [13] Orion: A holistic end-to-end autonomous driving framework by vision-language instructed action generation. arXiv:2503.19755, 2025.
>
> [14] Planning-oriented autonomous driving. In CVPR, 2023.
>
> [15] Drivetransformer: Unified transformer for scalable end-to-end autonomous driving. arXiv:2503.07656, 2025.

---

### Official Review · Reviewer_sdh6 · 2025-11-01

**Soundness:** 2
**Presentation:** 3
**Contribution:** 2
**Rating:** 4
**Confidence:** 5

**Summary:**

This paper proposes FlowAD, a novel modeling paradigm for autonomous driving that aims to address the lack of feedback from ego-motion to future observations in current models. The authors argue this gap leads to an incomplete understanding of the driving process. FlowAD's core idea is to model this "ego-scene interaction" as a "scene flow" relative to the ego-vehicle. The paper also introduces a new FCP metric.

**Strengths:**

1. Modeling the ego-scene interaction as "scene flow" in the feature space, inspired by human optic flow, is an interesting approach to the problem.
2. The paper is well written.

**Weaknesses:**

1. The newly proposed FCP metric judges "correctness" using L2 distance to the Ground Truth (GT) trajectory. This is a strong and potentially flawed assumption, as "closeness to GT" does not equal "correct" in planning. A safer, more conservative plan (e.g., braking earlier) might deviate from the GT but would be unfairly penalized by FCP as a "slow" or "wrong" response.
2. The paper does not demonstrate how FCP, an open-loop metric based on GT, translates to or predicts better closed-loop performance. Its value in a closed-loop context is not well-argued, only discuss in open-loop is some how meaningless.
3. The paper's SOTA claim is weakened by its comparison to relatively dated baselines (e.g., SparseDrive, UniAD). This makes it difficult to assess FlowAD's true performance.
4. The "scene flow" seems to primarily model the relative motion of the static background caused by ego-motion. It is unclear how this "ego-centric" flow model effectively handles the more critical, non-ego-centric dynamics, such as another vehicle suddenly cutting in while the ego-vehicle is driving straight.

**Questions:**

1. Can the authors provide analysis linking a better FCP score (open-loop) to a specific, measurable improvement in closed-loop driving (e.g., faster reaction to hazards), beyond the overall driving score?
2. Could the authors provide comparisons against more recent end-to-end planners (e.g., GenAD (ECCV24), DriveTransformer (ICLR25), MomAD (CVPR25), ORION (ICCV25), etc.) on both open-loop and closed-loop?

---

> ### Author Response · Authors · 2025-11-23
> **(Part1/3) Response to Reviewer #sdh6**
>
> **Q1:** The FCP's dependence on GT.
>
> **R1:** We appreciate this constructive suggestion. While reliance on Ground Truth (GT) is a standard practice for many open-loop metrics (e.g., L2 Error) to establish a baseline, we acknowledge that it cannot fully capture the intrinsic correctness or the multi-modal nature of human-like planning. To address this limitation and better evaluate reaction speed, we have extended the FCP metric to move beyond simple GT matching. Specifically, we define a ''successful reaction'' based on the trajectory's compliance with the high-level command rather than its proximity to the GT. For a given command (provided in both open- and closed-loop settings), the model is considered responsive if its predicted trajectory at the $3\text{s}$ horizon aligns with the command for a predefined sequence of $q$ frames. For instance, regarding a ''Turn Right'' command, adopting the threshold from UniAD [1], a planner is deemed successful if the lateral displacement of the ego trajectory at the final timestamp exceeds $2.0\text{m}$ for $q$ consecutive frames (please refer to Sec. A.12 of the appendix for the detailed formulation, thanks!)
>
> **Table 1: Planning results in terms of L2 error, collision rate, and the proposed FCP metrics with different thresholds on the nuScenes [2] validation set. The baseline method is SparseDrive [3]. $^{\diamondsuit}$ denotes the extended FCP metric.**
>
> | Method | Backbone | L2 1s | L2 2s | L2 3s | L2 Avg | Col 1s | Col 2s | Col 3s | Col Avg | FCP 0.25m | FCP 0.50m | FCP 0.75m | FCP Avg | FCP $^{\diamondsuit}$ 1frame | FCP $^{\diamondsuit}$ 2frame | FCP $^{\diamondsuit}$ 3frame | FCP $^{\diamondsuit}$ Avg | FPS |
> | :--- | :--- | :--- | :--- | :--- | :--- | :--- | :--- | :--- | :--- | :--- | :--- | :--- | :--- | :--- | :--- | :--- | :--- | :--- |
> | UniAD [1] | ResNet101 | 0.44 | 0.67 | 0.96 | 0.69 | 0.04 | 0.08 | 0.23 | 0.12 | 5.41 | 2.69 | 0.93 | 3.01 | 0.73 | 2.02 | 3.43 | 2.06 | 1.8 |
> | VAD [4] | ResNet50 | 0.41 | 0.70 | 1.05 | 0.72 | 0.07 | 0.17 | 0.41 | 0.22 | 6.44 | 3.07 | 1.42 | 3.64 | 0.87 | 2.39 | 4.12 | 2.46 | 5.2 |
> | MomAD [5] | ResNet50 | 0.31 | 0.57 | 0.91 | 0.60 | 0.01 | 0.05 | 0.22 | 0.09 | - | - | - | - | - | - | - | - | 7.8 |
> | SparseDrive [3] | ResNet50 | 0.29 | 0.58 | 0.96 | 0.61 | 0.01 | 0.05 | 0.18 | 0.08 | 4.87 | 2.55 | 1.09 | 2.84 | 0.54 | 1.88 | 3.02 | 1.48 | **9.0** |
> | **FlowAD (Ours)** | **ResNet50** | **0.28** | **0.55** | **0.84** | **0.56** | **0.01** | **0.03** | **0.15** | **0.06** | **2.12** | **1.03** | **0.48** | **1.21** | **0.37** | **0.90** | **1.79** | **1.02** | 7.6 |
>
> We then perform comparison using the extended FCP $^{\diamondsuit}$ metric, as shown in Table 1 above. The results indicate that our method consistently maintains its superiority over competing approaches across varying thresholds of $q$, validating the robustness of our planner. Moreover, by eliminating GT dependence and accommodating diverse decision-making behaviors, this refined metric offers the flexibility to be deployed in both open-loop and closed-loop settings. We hope FCP $^{\diamondsuit}$ serves as a standard, robust benchmark for a more holistic assessment of autonomous driving capabilities.
>
> We have added the above analysis and experiment in Sec. A.12 of the appendix. Again, thanks!
>
> **Q2:** FCP's value in closed-loop context.
>
> **R2:** Thanks for this innovative suggestion.
> As discussed in Q1-R1, the refined FCP metric assesses the reaction speed of the planner by judging the predicted trajectory if it accords with the high-level command for a few frames. It is GT-independent and flexible to be applied to closed-loop evaluation.

---

> ### Author Response · Authors · 2025-11-23
> **(Part2/3) Response to Reviewer #sdh6**
>
> **Q3:** Weakened SOTA claim due to dated baselines.
>
> **R3:** We appreciate the reviewer for this helpful comment.
> As suggested, we incorporate more recent methods for comparison and further apply our approach to DiffusionDrive [6]. The results are listed in Table 2 below. Our planner still demonstrates superior performance with the proposed ego-scene interactive modeling, providing its effectiveness for better understanding of the driving process. We have updated Tab. 4 of the manuscript by adding the above-mentioned methods, as well as recent SOTAs, thanks!
>
> **Table 2: Results of perception, motion prediction and planning on the nuScenes validation set.** The FPS is measured on an RTX3090 GPU. The baselines are SparseDrive and DiffusionDrive. $^{\ddagger}$ denotes using ego status in the planning module.
>
> | Method | Backbone | Det mAP↑ | Det NDS↑ | Trk AMOTA↑ | Trk AMOTP↓ | Map mAP↑ | Mot minADE↓ | Mot minFDE↓ | Plan Avg.L2↓ | Plan Avg.Col↓ | Plan FCP↓ | FPS↑ |
> | :--- | :--- | :--- | :--- | :--- | :--- | :--- | :--- | :--- | :--- | :--- | :--- | :--- |
> | VAD [4] | ResNet50 | - | - | - | - | 0.476 | - | - | 0.72 | 0.21 | 3.07 | 5.2 |
> | MomAD [5] | ResNet50 | 0.423 | 0.531 | 0.391 | 1.243 | 0.559 | 0.61 | 0.98 | 0.60 | 0.09 | - | 7.8 |
> | DriveWM [7] | - | - | - | - | - | - | - | - | 0.80 | 0.26 | - | - |
> | LAW [8] | - | - | - | - | - | - | - | - | 0.61 | 0.30 | - | - |
> | UncAD [9] | ResNet50 | - | - | - | - | - | - | - | 0.60 | 0.07 | - | - |
> | FocalAD [10] | ResNet50 | - | - | - | - | - | 0.61 | 0.95 | 0.60 | 0.09 | - | - |
> | GenAD [11] | ResNet50 | 0.290 | - | - | - | - | - | - | 0.58 | 0.16 | 1.78 | 6.7 |
> | ORION (VLM) [12] | - | - | - | - | - | - | - | - | 0.63 | 0.37 | - | - |
> | SparseDrive [3] | ResNet50 | 0.418 | 0.525 | 0.386 | 1.254 | 0.551 | 0.62 | 0.99 | 0.61 | 0.08 | 2.55 | **9.0** |
> | **FlowAD $_{SparseDrive}$ (Ours)** | **ResNet50** | 0.448 | 0.555 | 0.427 | 1.181 | 0.582 | **0.58** | **0.95** | 0.56 | 0.06 | **1.03** | 7.6 |
> | DiffusionDrive [6] | ResNet50 | 0.412 | 0.522 | 0.374 | 1.246 | 0.559 | 0.64 | 1.01 | 0.57 | 0.08 | 1.35 | 6.8 |
> | **FlowAD $_{DiffusionDrive}$ (Ours)** | **ResNet50** | **0.477** | **0.567** | **0.466** | **1.141** | **0.605** | 0.61 | 0.97 | **0.54** | **0.04** | 1.05 | 5.4 |
> | UniAD [1] | ResNet101 | 0.380 | 0.498 | 0.359 | 1.320 | - | 0.71 | 1.02 | 0.69 | 0.12 | 2.69 | 1.8 |
> | SparseDrive [3] | ResNet101 | 0.496 | 0.588 | 0.501 | 1.085 | 0.562 | 0.60 | 0.96 | 0.58 | 0.06 | 2.30 | **7.3** |
> | **FlowAD $_{SparseDrive}$ (Ours)** | **ResNet101** | **0.523** | **0.605** | **0.518** | **1.040** | **0.595** | **0.56** | **0.93** | **0.52** | **0.05** | **0.91** | 5.4 |
> | DriveDreamer $^{\ddagger}$ [13] | - | - | - | - | - | - | - | - | 0.29 | 0.15 | - | - |
> | SSR $^{\ddagger}$ [14] | ResNet50 | - | - | - | - | - | - | - | 0.39 | 0.06 | 1.91 | **19.6** |
> | DriveTransformer $^{\ddagger}$ [15]  | ResNet50 | - | - | - | - | - | - | - | 0.40 | 0.11 | - | - |
> | **FlowAD $_{SparseDrive}^{\ddagger}$ (Ours)** | **ResNet50** | **0.451** | **0.554** | **0.430** | **1.198** | **0.576** | **0.57** | **0.94** | **0.25** | **0.04** | **0.75** | 7.6 |

---

> ### Author Response · Authors · 2025-11-23
> **(Part3/3) Response to Reviewer #sdh6**
>
> **Q4:** FlowAD's performance in non-ego-centric dynamic scenarios.
>
> **R4:** Thanks for the helpful and innovative suggestion.
> FlowAD's scene flow modeling captures object dynamics beyond static background through local aggregation that handles object fragmentation. Therefore, our approach essentially models the interactive dynamics between the vehicle, the static environment, and moving objects. This enhances the model’s understanding of autonomous driving and its capacity to capture scene dynamics. Consequently, our method can handle non-ego-centric dynamics, such as the "cut-in" scenario. We further demonstrate this by the case in Fig. 9 of the appendix. As a leading vehicle performs a cut-in, our planner has already inferred its motion state from historical observations and produces a salient response, corroborating the above hypothesis. Besides, the superiority of the ability metrics in Bench2Drive [16] (e.g., Merging and Emergency Brake that include many cutting-in scenarios, Tab. 5 of the manuscript) substantiates our method’s advantage in modeling vehicles that are not centered on ego-motion.
>
> We have added the above analysis and visualizations in Sec. A.18 of the appendix, thanks!
>
> ## References
> [1] Planning-oriented autonomous driving. In CVPR, 2023.
>
> [2] Nuscenes: A multimodal dataset for autonomous driving. In CVPR, 2020.
>
> [3] Sparsedrive: End-to-end autonomous driving via sparse scene representation. arXiv:2405.19620, 2024.
>
> [4] Vad: Vectorized scene representation for efficient autonomous driving. arXiv:2303.12077, 2023.
>
> [5] Don’t shake the wheel: Momentum-aware planning in end-to-end autonomous driving. arXiv:2503.03125, 2025.
>
> [6] Diffusiondrive: Truncated diffusion model for end-to-end autonomous driving. In CVPR, 2025.
>
> [7] Driving into the future: Multiview visual forecasting and planning with world model for autonomous driving. In CVPR, 2024.
>
> [8] Enhancing end-to-end autonomous driving with latent world model. arXiv:2406.08481, 2024.
>
> [9] Uncad: Towards safe end-to-end autonomous driving via online map uncertainty. arXiv:2504.12826, 2025.
>
> [10] Focalad: Local motion planning for end-to-end autonomous driving. arXiv:2506.11419, 2025.
>
> [11] Genad: Generative end-to-end autonomous driving. In ECCV, 2024.
>
> [12] Orion: A holistic end-to-end autonomous driving framework by vision-language instructed action generation. arXiv:2503.19755, 2025.
>
> [13] Drivedreamer: Towards real-world-drive world models for autonomous driving. In ECCV, 2024.
>
> [14] Navigation-guided sparse scene representation for end-to-end autonomous driving. arXiv:2409.18341, 2024.
>
> [15] Drivetransformer: Unified transformer for scalable end-to-end autonomous driving. arXiv:2503.07656, 2025.
>
> [16] Bench2drive: Towards multi-ability benchmarking of closed-loop end-to-end autonomous driving. arXiv:2406.03877, 2024.

---

> ### Author Response · Authors · 2025-11-26
>
> Dear Reviewer sdh6, Thank you once again for your valuable comments on our submission. As the discussion phase is approaching its end, we would like to kindly confirm whether we have sufficiently addressed all of your concerns (or at least part of them). Should there be any remaining questions or areas requiring further clarification, please do not hesitate to let us know. If you are satisfied with our responses, we would greatly appreciate your consideration in adjusting the evaluation scores accordingly. We sincerely look forward to your feedback.

---

### Author Response · Authors · 2025-12-01
**Important Context Regarding Score Rollback and Rebuttal Status**

Dear Area Chair,

Following the official score rollback caused by the recent incident, we are writing to ensure you are fully informed that our paper previously received strong positive ratings from three reviewers before the reset. We would like to highlight that during the rebuttal phase reviewer n4vr explicitly affirmed our submission as excellent work and maintained their original score of 8, whereas reviewer sdh6 who assigned a score of 4 (the only negative score) remained unresponsive despite the comprehensive supplementary experiments and detailed discussions we provided to address their concerns. Given that we are now unable to engage further with this reviewer, we kindly request that you consider this context and the effectiveness of our rebuttal in your final assessment.

---

### Meta-Review · Area_Chair_Apr6 · 2026-01-05

**Summary:**

The paper proposes FlowAD, a framework for autonomous driving that explicitly models the feedback of ego-motion on scene observations. To address the limitation that current paradigms often neglect the interaction between ego-motion and future observations, the paper introduces an "ego-scene interactive modeling" paradigm. The core method involves an Ego-Guided Scene Partition to construct flow units based on steering and velocity, followed by Spatiotemporal Flow Prediction to model dynamics. Additionally, a new metric named FCP is proposed to assess scene understanding.

**Reviewer Concerns:**

Validity of the FCP Metric: Reviewers sdh6 and n4vr questioned the original FCP metric's reliance on Ground Truth (GT), arguing that "closeness to GT" does not equal "correct planning". The paper responded by proposing an extended FCP metric that is GT-independent. This refined version evaluates whether the trajectory complies with high-level commands (e.g., lateral displacement thresholds) rather than simple L2 distance, making it applicable to closed-loop settings.

Baselines and State-of-the-Art: Reviewers sdh6 and 8pZR noted that the original comparisons relied on dated baselines (e.g., SparseDrive, UniAD). In the rebuttal, the paper added comprehensive comparisons against recent methods, including GenAD, Drive Transformer, Diffusion Drive, SSR, and LAW. FlowAD maintained its superiority even against these newer perception-free and diffusion-based approaches.

Robustness and Sensitivity: Reviewers 8pZR and 17iF raised concerns about the method's sensitivity to noisy ego-pose estimation and partition parameters. The paper provided ablation studies with Gaussian noise injections, calibration errors, and temporal drift, demonstrating that FlowAD is robust.

Efficiency (FPS): Reviewer 17iF noted a drop in inference speed (9.0 FPS vs 7.6 FPS). The paper demonstrated a "lightweight" configuration that applies modeling only to the last feature stage, recovering speed to 8.5 FPS with negligible performance loss.

**Reviewer Scores:**

Reviewer n4vr: 8
Reviewer 17iF: 8
Reviewer 8pZR: 6
Reviewer sdh6: 4

---

### Decision · Program_Chairs · 2026-01-26

Accept (Poster)